# A simple connection from loss flatness to compressed neural representations

**Shirui Chen**                                                                 *sc256@uw.edu*
*Department of Applied Mathematics*
*Computational Neuroscience Center*
*University of Washington*

**Stefano Recanatesi**                                           *stefano.recanatesi@gmail.com*
*Technion, Israel Institute of Technology*

**Eric Shea-Brown**                                                            *etsb@uw.edu*
*Department of Applied Mathematics*
*Computational Neuroscience Center*
*University of Washington*

**Reviewed on OpenReview:** *https://openreview.net/forum?id=GgpQbU9bFR*

## Abstract

Despite extensive study, the fundamental significance of sharpness—the trace of the loss Hessian at local minima—remains unclear. While often associated with generalization, recent work reveals inconsistencies in this relationship. We explore an alternative perspective by investigating how sharpness relates to the geometric structure of neural representations in feature space. Specifically, we build from earlier work by Ma and Ying to broadly study compression of representations, defined as the degree to which neural activations concentrate when inputs are locally perturbed. We introduce three quantitative measures: the Local Volumetric Ratio (LVR), which captures volume contraction through the network; the Maximum Local Sensitivity (MLS), which measures maximum output change normalized by the magnitude of input perturbations; and Local Dimensionality, which captures uniformity of compression across directions.

We derive upper bounds showing that LVR and MLS are mathematically constrained by sharpness: flatter minima necessarily limit these compression metrics. These bounds extend to reparametrization-invariant sharpness (measures unchanged under layer rescaling), addressing a key limitation of standard sharpness. We introduce network-wide variants (NMLS, NVR) that account for all layer weights, providing tighter and more stable bounds than prior single-layer analyses. Empirically, we validate these predictions across feedforward, convolutional, and transformer architectures, demonstrating consistent positive correlation between sharpness and compression metrics. Our results suggest that sharpness fundamentally quantifies representation compression rather than generalization directly, offering a resolution to contradictory findings on the sharpness-generalization relationship and establishing a principled mathematical link between parameter-space geometry and feature-space structure. Code is available at `https://github.com/chinsengi/sharpness-compression`.

## 1 Introduction

There has been long-standing interest in sharpness, a geometric metric in the *parameter* space that measures the flatness of the loss landscape at local minima. Flat minima refer to regions in the loss landscape where the loss function has a relatively large basin, and the loss does not change much in different directions around the minimum. Empirical studies and theoretical analyses have shown that training deep neural

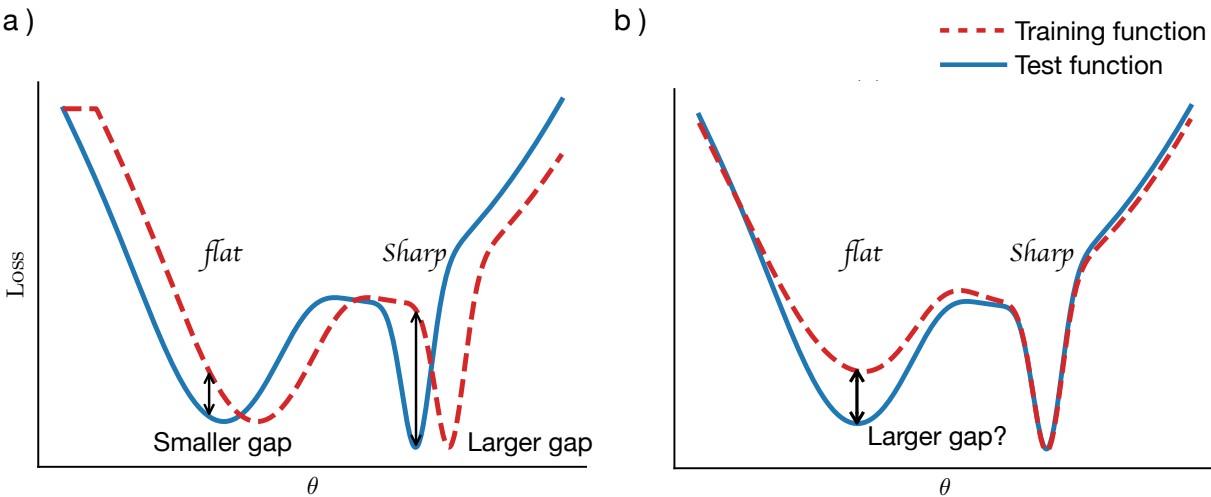

Figure 1: A schematic illustration of why flatness alone does not reliably predict generalization. **(a)** Under the common but unjustified assumption that the test loss is a horizontal shift of the training loss, flat minima trivially exhibit a smaller generalization gap than sharp minima. **(b)** In a hypothetical but possible scenario, the test loss differs from the training loss by an additive perturbation rather than a simple shift. Here the flat minimum can exhibit a *larger* generalization gap than the sharp minimum, demonstrating that curvature alone is insufficient to predict generalization.

networks using stochastic gradient descent (SGD) with a small batch size and a high learning rate often converges to flat and wide minima (Ma & Ying, 2021; Blanc et al., 2020b; Geiger et al., 2021; Li et al., 2022; Wu et al., 2018; Jastrzebski et al., 2018; Xie et al., 2021; Zhu et al., 2019). Many works conjecture that flat minima lead to a simpler model (shorter description length), and thus are less likely to overfit and more likely to generalize well (Hochreiter & Schmidhuber, 1997; Keskar et al., 2016; Wu et al., 2018; Yang et al., 2023). Based on this rationale, sharpness-aware minimization (SAM) has been a popular method for improving a model's generalization ability. However, recent work has shown that SAM does not *only* minimize sharpness to achieve superior generalization performance (Andriushchenko & Flammarion, 2022; Wen et al., 2023). More confusingly, it remains unclear whether flatness correlates positively with the generalization capacity of the network (Dinh et al., 2017; Andriushchenko et al., 2023; Yang et al., 2021), and even when it does, the correlation is not perfect (Neyshabur et al., 2017; Jiang et al., 2019). In particular, Dinh et al. (2017) argues that one can construct very sharp networks that generalize well through reparametrization, while Andriushchenko et al. (2023) show that even reparametrization-invariant sharpness cannot capture the relationship between sharpness and generalization.

Figure 1 illustrates the crux of this issue. The common intuition for why flat minima should generalize better (Hochreiter & Schmidhuber, 1997; Keskar et al., 2016) implicitly assumes that the test loss landscape is a horizontally shifted copy of the training loss (Figure 1a): under such a shift, the loss at a flat minimum changes little while the loss at a sharp minimum increases substantially, yielding a smaller generalization gap at the flat minimum. However, there is no principled justification for this horizontal-shift assumption. In a hypothetical setting where the test loss differs from the training loss by a random perturbation (Figure 1b), the generalization gap at a flat minimum could be *larger* than at a sharp minimum, contradicting the standard

narrative. This suggests that sharpness, while clearly capturing meaningful geometric structure, may not be directly measuring generalization ability.

In light of this paradox, we take a different perspective: rather than asking what sharpness tells us about generalization, we ask what sharpness tells us about the *representations* learned by the network. We show that there exists a more consistent relationship by investigating how sharpness near interpolation solutions in the *parameter* space influences local geometric features of neural representations in the *feature* space. By building a relationship between sharpness and the local compression of neural representations, we argue that sharpness, in its essence, measures the compression of neural representations. Specifically, we show that as sharpness decreases and the minimum flattens, sharpness-related quantities upper-bound certain compression measures, meaning that the neural representation must also undergo some degree of compression. We also note how local dimensionality is a compression metric of a distinct nature and therefore does not necessarily correlate with sharpness.

More specifically, our work makes the following novel contributions:

1. **Refined and new compression metrics with sharpness bounds.** We refine and identify feature space quantities that quantify compression and are bounded by sharpness: local volumetric ratio (LVR) and maximum local sensitivity (MLS). The former is a new metric in this context and the latter refines a related quantity in earlier work. [1] We derive explicit upper bounds showing these metrics are constrained by sharpness-related quantities. This establishes a direct mathematical link between parameter-space geometry (sharpness) and feature-space geometry (compression), applicable to feedforward networks with quadratic loss.

2. **Improved bounds via network-wide metrics.** We improve the bound on MLS from Ma & Ying (2021) and propose Network MLS (NMLS), which considers sensitivity at all intermediate layers rather than only the input. This yields bounds that consistently predict positive correlation with sharpness across diverse experimental settings, overcoming limitations of single-layer analyses where prefactors can dominate and obscure the sharpness-compression relationship.

3. **Extension to reparametrization-invariant sharpness.** We prove that using reparametrization-invariant sharpness (i.e., unchanged under layer rescaling) tightens our bounds, addressing a key criticism of standard sharpness measures and providing a novel interpretation: reparametrization-invariant sharpness quantifies robustness of outputs to internal representations.

4. **Empirical validation across architectures.** We validate our theoretical predictions through experiments on feedforward (MLP), convolutional (VGG-11, LeNet), and transformer (ViT) architectures, demonstrating that LVR and MLS/NMLS consistently correlate with their sharpness-related bounds during training and across pretrained models.

5. **Connection to neural collapse and intermediate representations.** We extend our analysis to penultimate and intermediate layer representations, relating our framework to neural collapse phenomena. Our bounds apply most naturally when the number of classes exceeds feature dimension, relevant for settings like language modeling and large-scale retrieval.

**On bound tightness.** We emphasize that our theoretical results provide *upper bounds* rather than equalities. Moreover, these bounds depend not only on sharpness but also on weight norms and input magnitudes. Consequently, for two networks with identical sharpness but different weight scales, the bounds may differ. However, our extensive empirical validation demonstrates that despite this apparent looseness, the bounds are sufficient to predict consistent positive correlation between sharpness and compression metrics across training dynamics and diverse architectures. This empirical validation is essential to our claims and demonstrates the practical utility of our theoretical framework.

With these results, we help reveal the nature of sharpness through the interplay between key properties of trained neural networks in parameter space and feature space.

---

[1]We collectively term MLS, NMLS as "compression metrics", because these quantities measure how compressed/concentrated a set of noise-perturbed input/internal neural representations is after going through the network.

**Paper organization.** Section 2 reviews the pioneering arguments of Ma & Ying (2021) that flatter minima can constrain the gradient of network output with respect to network input, and extends the formulation to the multidimensional input case. Section 3 proves that lower sharpness implies a lower upper bound on two metrics of the compression of the representation manifold in feature space: the local volumetric ratio and the maximum local sensitivity (MLS) (Section 3.1, Section 3.2). Section 4 empirically verifies our theory by calculating various compression metrics, their theoretical bounds, and sharpness for models during training as well as pretrained ones. Finally, Section 5 discusses how these conditions help explain why there are mixed results on the relationship between sharpness and generalization in the literature, by looking through the alternative lens of compressed neural representations.

**Key terminology.** Before proceeding, we clarify key terms used throughout this work. *Reparametrization* refers to equivalent network parameterizations that produce identical input-output mappings; for example, scaling the weights of layer $l$ by factor $\alpha$ and layer $l + 1$ by $1/\alpha$ leaves the network function with ReLU nonlinearity unchanged as demonstrated in Dinh et al. (2017). Reparametrization-invariant sharpness measures remain constant under such transformations, addressing the concern that standard sharpness can be arbitrarily changed without affecting network behavior.

The compression metrics we study are: *Local Volumetric Ratio (LVR)*, which quantifies how much a small input volume contracts as it propagates through the network; *Maximum Local Sensitivity (MLS)*, which measures the maximum rate of change of network output with respect to input perturbations; and *Network MLS (NMLS)*, which extends MLS by considering sensitivity at all intermediate layer representations rather than only the input layer.

## 2 Background and setup

Consider a feedforward neural network $f$ with input data $\mathbf{x} \in \mathbb{R}^M$ and parameters $\boldsymbol{\theta}$. The output of the network is:

$$\mathbf{y} = f(\mathbf{x}; \boldsymbol{\theta}) \ , \tag{1}$$

where $\mathbf{y} \in \mathbb{R}^N$ ($N < M$). We consider a quadratic loss $L(\mathbf{y}, \mathbf{y}_{\text{true}}) = \frac{1}{2} \|\mathbf{y} - \mathbf{y}_{\text{true}}\|^2$, a function of the outputs and ground truth $\mathbf{y}_{\text{true}}$. In the following, we will simply write $L(\mathbf{y})$, $L(f(\mathbf{x}, \boldsymbol{\theta}))$ or simply $L(\boldsymbol{\theta})$ to highlight the dependence of the loss on the output, the network, or its parameters.

Sharpness measures how much the loss gradient changes when the network parameters are perturbed, and is defined by the sum of the eigenvalues of the Hessian:

$$S(\boldsymbol{\theta}) = \text{Tr}(H) \ , \tag{2}$$

with $H = \nabla^2 L(\boldsymbol{\theta})$ being the Hessian. The trace of the Hessian, $\text{Tr}\left(\nabla^2 L(\theta)\right)$, is not the only definition of sharpness, but many sharpness minimization methods have been theoretically shown to reduce this quantity in interpolating models. Specifically, assuming that the training loss minimizers lie on a smooth manifold (Cooper, 2018; Fehrman et al., 2020), methods like Sharpness-Aware Minimization (SAM) (Foret et al., 2020) when used with batch size 1 and sufficiently small learning rate and perturbation radius (Wen et al., 2022; Bartlett et al., 2023), or Label Noise SGD with a small enough learning rate (Blanc et al., 2020a; Damian et al., 2021), tend to favor interpolating solutions with a low Hessian trace. Therefore, we focus our analysis on the trace of the Hessian.

**Scope and loss function considerations.** Our theoretical analysis is conducted using the quadratic (MSE) loss, which allows us to derive the key relationship in Equation (3) connecting sharpness to network gradients. However, we note important considerations regarding other loss functions. For cross-entropy (CE) loss, Granziol (2020) showed that Hessian-based flatness measures become unreliable: solutions with large weights and low CE loss can exhibit artificially small Hessian traces, appearing deceptively "flat" despite poor generalization properties. Moreover, the Hessian rank of CE loss is bounded by the number of neurons times the number of classes, causing many eigenvalues to be zero or near-zero regardless of the solution quality. Consequently, networks with L2 regularization (which have smaller weights) can be sharper yet generalize

better than unregularized networks, directly contradicting the flatness-generalization hypothesis for CE loss. This fundamentally breaks the connection between CE loss sharpness and generalization.

Despite this limitation of CE loss sharpness, our framework can still provide insights for networks trained with CE loss through two pathways. First, our theoretical results extend to logistic loss with label smoothing because using the logistic loss with label smoothing yields the same set of minimizers and flattest minimizers as a corresponding problem using mean squared error (see Lemma A.13 in Wen et al. (2023)). Second, and more practically relevant, our compression metrics (LVR, MLS, NMLS) depend on the network function $f$ itself rather than the training loss. Once a network is trained (with any loss function), we can analyze its compression properties by examining how it transforms inputs to outputs. The empirical success of this approach is demonstrated in Figure 5, where we observe meaningful correlations between compression and sharpness (computed assuming MSE loss) for 181 pretrained ViT models that were trained with CE loss. For these networks, we compute sharpness using MSE loss applied to the trained network, which allows us to differentiate between solutions even though their original CE training loss was near zero.

Following (Ma & Ying, 2021; Ratzon et al., 2023), we define $\boldsymbol{\theta}^*$ to be an "exact interpolation solution" on the zero training loss manifold in the parameter space (the zero loss manifold in what follows), where $f(\mathbf{x}_i, \boldsymbol{\theta}^*) = \mathbf{y}_i$ for all $i$'s (with $i \in \{1..n\}$ indexing the training set) and $L(\boldsymbol{\theta}^*) = 0$. On the zero loss manifold, in particular, we have

$$S(\boldsymbol{\theta}^*) = \frac{1}{n} \sum_{i=1}^{n} \|\nabla_{\boldsymbol{\theta}} f(\mathbf{x}_i, \boldsymbol{\theta}^*)\|_F^2, \tag{3}$$

where $\|\cdot\|_F$ is the Frobenius norm. We state a proof of this equality, which appears in Ma & Ying (2021) and Wen et al. (2023), in Appendix C. In practice, the parameter $\boldsymbol{\theta}$ will never reach an exact interpolation solution due to the gradient noise of SGD; however, Equation (3) is a good approximation of the sharpness when the training loss is sufficiently small (see error bounds in Lemma C.1).

To see why minimizing the sharpness of the solution leads to more compressed representations, we need to move from the parameter space to the input space. To do so we clarify the proof of Equation (4) in Ma & Ying (2021) that relates adversarial robustness to sharpness in the following. The improvements we made are summarized at the end of this section. Let $\mathbf{W}$ be the input weights (the parameters of the first linear layer) of the network, and $\bar{\boldsymbol{\theta}}$ be the rest of the parameters. Following (Ma & Ying, 2021), as the weights $\mathbf{W}$ multiply the inputs $\mathbf{x}$, we have the following identities:

$$\|\nabla_{\mathbf{W}} f(\mathbf{W}\mathbf{x}; \bar{\boldsymbol{\theta}})\|_F = \sqrt{\sum_{i,j,k} J_{jk}^2 x_i^2} = \|J\|_F \|\mathbf{x}\|_2 \geq \|J\|_2 \|\mathbf{x}\|_2 ,$$
$$\nabla_{\mathbf{x}} f(\mathbf{W}\mathbf{x}; \bar{\boldsymbol{\theta}}) = J\mathbf{W} , \tag{4}$$

where $J = \frac{\partial f(\mathbf{W}\mathbf{x}; \bar{\boldsymbol{\theta}})}{\partial (\mathbf{W}\mathbf{x})}$ is a complex expression computed with backpropagation. From Equation (4) and the sub-multiplicative property of the Frobenius norm and the matrix 2-norm [2], we have:

$$\|\nabla_{\mathbf{x}} f(\mathbf{W}\mathbf{x}; \bar{\boldsymbol{\theta}})\|_2 \leq \|\nabla_{\mathbf{x}} f(\mathbf{W}\mathbf{x}; \bar{\boldsymbol{\theta}})\|_F \leq \frac{\|\mathbf{W}\|_2}{\|\mathbf{x}\|_2} \|\nabla_{\mathbf{W}} f(\mathbf{W}\mathbf{x}; \bar{\boldsymbol{\theta}})\|_F . \tag{5}$$

We call Equation (5) the linear stability trick. As a result, we have

$$\frac{1}{n} \sum_{i=1}^{n} \|\nabla_{\mathbf{x}} f(\mathbf{x}_i, \boldsymbol{\theta}^*)\|_2^k \leq \frac{1}{n} \sum_{i=1}^{n} \|\nabla_{\mathbf{x}} f(\mathbf{x}_i, \boldsymbol{\theta}^*)\|_F^k$$
$$\leq \frac{\|\mathbf{W}\|_2^k}{\min_i \|\mathbf{x}_i\|_2^k} \frac{1}{n} \sum_{i=1}^{n} \|\nabla_{\mathbf{W}} f(\mathbf{x}_i, \boldsymbol{\theta}^*)\|_F^k \tag{6}$$
$$\leq \frac{\|\mathbf{W}\|_2^k}{\min_i \|\mathbf{x}_i\|_2^k} \frac{1}{n} \sum_{i=1}^{n} \|\nabla_{\boldsymbol{\theta}} f(\mathbf{x}_i, \boldsymbol{\theta}^*)\|_F^k.$$

---

[2] $\|AB\|_F \leq \|A\|_F \|B\|_2$, $\|AB\|_2 \leq \|A\|_2 \|B\|_2$

This reveals the impact of flatness on the input sensitivity when $k = 2$. Equation (6) holds for any positive $k$. Thus, the effect of input perturbations is upper-bounded by the sharpness of the loss function (cf. Equation (3)). Note that Equation (6) corresponds to Equation (4) in Ma & Ying (2021) with multivariable output.

While the experiments of Ma & Ying (2021) *empirically* show a high correlation between the left-hand side of Equation (6) and the sharpness, Equation (6) does not explain such a correlation by itself because of the scaling factor $\|\mathbf{W}\|_2^k / \min_i \|\mathbf{x}_i\|_2^k$. This factor makes the right-hand side of Equation (6) highly variable, leading to mixed positive and/or negative correlations with sharpness under different experimental settings. In the next section, we will improve this bound to relate sharpness to various metrics measuring robustness and compression of representations. More specifically, compared to Equation (4) of Ma & Ying (2021), we make the following improvements:

1. We replace the reciprocal of the minimum with the quadratic mean to achieve a more stable bound (Proposition 3.8). This term remains relevant as common practice in deep learning does *not* normalize the input by its 2-norm, as this would erase information about the modulus of the input.

2. While Ma & Ying (2021) only considers scalar output, we extend the result to consider networks with multivariable output throughout the paper.

3. We introduce new metrics such as Network Volumetric Ratio and Network MLS (Definition 3.5 and Definition 3.9) and their sharpness-related bounds (Proposition 3.6 and Proposition 3.10), which have two advantages compared to prior results (cf. the right-hand side of Equation (6)):

   (a) our metrics consider all linear weights so that bounds remain stable to weight changes during training.
   (b) they avoid the gap between derivative w.r.t. the first layer weights and the derivative w.r.t. all weights, i.e. the second inequality in Eq. 6, thus tightening the bound.

Moreover, we show that the underlying theory readily extends to networks with residual connections in Appendix B.

## 3 From robustness to inputs to compression of representations

Building upon the linear stability bound (Equation (5)) established in Section 2, this section derives our main theoretical results connecting sharpness to compression of neural representations.

**Key assumptions.** Our analysis assumes: (i) the network has reached an approximate interpolation solution $\boldsymbol{\theta}^*$ where training loss is near zero, allowing us to use the sharpness characterization from Equation (3); (ii) the network architecture consists of linear layers (including convolutional layers, which are linear operations) with nonlinear activations, though we do not assume specific activation functions for the main bounds (the theory extends to networks with residual/skip connections as detailed in Appendix B); and (iii) all bounds are derived using the quadratic (MSE) loss function. While our theoretical derivations require MSE loss, the compression metrics themselves (LVR, MLS, NMLS) depend only on the network function $f$ and can be computed for networks trained with any loss function, as discussed in Section 2. Throughout, we work in the local regime where first-order Taylor approximations are valid.

### 3.1 Sharpness bounds local volumetric transformation in the feature space

To understand how networks compress their representations, we first study how local volumes in input space transform as they pass through the network. If a network compresses its inputs, small volumes around input points should contract as they are mapped to feature space. We quantify this via the local volumetric ratio, the ratio between the volume of a hypercube of side length $h$ at $\mathbf{x}$, $H(\mathbf{x})$, and its image under transformation $f$, $f(H(\mathbf{x}), \boldsymbol{\theta}^*)$. By the change of variables formula from multivariate calculus, the infinitesimal volume

transformation is characterized by the Jacobian matrix. Consider the mapping $f : \mathbb{R}^M \to \mathbb{R}^N$ with $N < M$ and Jacobian $J = \nabla_{\mathbf{x}} f \in \mathbb{R}^{N \times M}$. The volume scaling factor is:

$$
\begin{aligned}
d\,\mathrm{Vol}|_{f(\mathbf{x}, \boldsymbol{\theta}^*)} &= \lim_{h \to 0} \frac{\mathrm{Vol}(f(H(\mathbf{x}), \boldsymbol{\theta}^*))}{\mathrm{Vol}(H(\mathbf{x}))} \\
&= \sqrt{\det(JJ^T)} \quad \text{(product of singular values of } J) \\
&= \sqrt{\det\left(\nabla_{\mathbf{x}} f (\nabla_{\mathbf{x}} f)^T\right)} ,
\end{aligned}
\tag{7}
$$

where the equality holds because for a rectangular matrix $J \in \mathbb{R}^{N \times M}$ with $N < M$, the $N$-dimensional volume scaling equals the product of its $N$ singular values $\sigma_1, \ldots, \sigma_N$, which equals $\sqrt{\det(JJ^T)}$ since $JJ^T \in \mathbb{R}^{N \times N}$ is a square matrix with eigenvalues $\sigma_1^2, \ldots, \sigma_N^2$. We formalize this as a definition:

**Definition 3.1.** *The **Local Volumetric Ratio at input** $\mathbf{x}$ of a network $f$ with parameters $\boldsymbol{\theta}$ is defined as $d\,\mathrm{Vol}|_{f(\mathbf{x}, \boldsymbol{\theta})} = \sqrt{\det\left(\nabla_{\mathbf{x}} f (\nabla_{\mathbf{x}} f)^T\right)}$.*

Exploiting the bound on the gradients derived earlier in Equation (5), we derive a similar bound for the volumetric ratio:

**Lemma 3.2.**

$$
d\,\mathrm{Vol}|_{f(\mathbf{x}, \boldsymbol{\theta}^*)} \leq \left( \frac{\mathrm{Tr}\, \nabla_{\mathbf{x}} f (\nabla_{\mathbf{x}} f)^T}{N} \right)^{N/2} = N^{-N/2} \|\nabla_{\mathbf{x}} f(\mathbf{x}, \boldsymbol{\theta}^*)\|_F^N ,
\tag{8}
$$

*Proof.* The inequality follows from the arithmetic-geometric mean inequality applied to the squared singular values of $\nabla_{\mathbf{x}} f$. The equality uses the trace identity $\mathrm{Tr}(\nabla_{\mathbf{x}} f (\nabla_{\mathbf{x}} f)^T) = \|\nabla_{\mathbf{x}} f\|_F^2$. See Appendix D for the complete derivation. $\square$

Next, we introduce a measure of the volumetric ratio averaged across input samples.

**Definition 3.3.** *The **Local Volumetric Ratio (LVR)** of a network $f$ with parameters $\boldsymbol{\theta}$ is defined as the sample mean of Local Volumetric Ratio at different input samples: $dV_{f(\boldsymbol{\theta})} = \frac{1}{n} \sum_{i=1}^{n} d\,\mathrm{Vol}|_{f(\mathbf{x}_i, \boldsymbol{\theta})}$.*

We now establish our first main result: an upper bound on LVR in terms of sharpness. This bound will show that flatter minima (lower sharpness) necessarily constrain the volumetric ratio, establishing a mathematical link between parameter-space flatness and feature-space compression.

**Proposition 3.4.** *The local volumetric ratio is upper bounded by a sharpness related quantity:*

$$
dV_{f(\boldsymbol{\theta}^*)} \leq \frac{N^{-N/2}}{n} \sum_{i=1}^{n} \|\nabla_{\mathbf{x}} f(\mathbf{x}_i, \boldsymbol{\theta}^*)\|_F^N \leq \frac{1}{n} \sqrt{\sum_{i=1}^{n} \frac{\|\mathbf{W}\|_2^{2N}}{\|\mathbf{x}_i\|_2^{2N}} \left( \frac{nS(\boldsymbol{\theta}^*)}{N} \right)^{N/2}}
\tag{9}
$$

*for all $N \geq 1$.*

The proof of the above inequalities is given in Appendix D.

The bound in Proposition 3.4 only considers sensitivity at the input layer. However, we can strengthen our analysis by examining compression at all intermediate layers, which allows us to incorporate all network weights rather than just the first-layer weights. This yields tighter bounds that are more stable during training. We denote the input to the $l$-th linear layer as $\mathbf{x}_i^l$ for $l = 1, 2, \cdots, L$, where $\mathbf{x}_i^1 = \mathbf{x}_i$ is the network input. Similarly, $\mathbf{W}_l$ is the weight matrix of the $l$-th linear/convolutional layer. With a slight abuse of notation, we use $f_l$ to denote the mapping from the input of the $l$-th layer to the final output.

**Definition 3.5.** *The **Network Volumetric Ratio (NVR)** is defined as the sum of the local volumetric ratios $dV_{f_l}$ for all $f_l$, that is, $dV_{net} = \sum_{l=1}^{L} dV_{f_l}$*

By applying Equation (9) to every intermediate layer, we obtain a network-wide bound:

**Proposition 3.6.** *The network volumetric ratio is upper bounded by a sharpness related quantity:*

$$\sum_{l=1}^{L} dV_{f_l} \leq \frac{N^{-N/2}}{n} \sum_{l=1}^{L} \sum_{i=1}^{n} \|\nabla_{\mathbf{x}^l} f_i^l\|_F^N \leq \frac{1}{n} \sqrt{\sum_{l=1}^{L} \sum_{i=1}^{n} \frac{\|\mathbf{W}_l\|_2^{2N}}{\|\mathbf{x}_i^l\|_2^{2N}} \cdot \left(\frac{nS(\boldsymbol{\theta}^*)}{N}\right)^{N/2}} . \tag{10}$$

Again, a detailed derivation of the above inequalities is given in Appendix D. Proposition 3.4 and Proposition 3.6 imply that flatter minima of the loss function in parameter space contribute to local compression of the data's representation manifold.

## 3.2 Maximum Local Sensitivity as an allied metric to track neural representation geometry

We observe that the equality condition in the first line of Equation (8) rarely holds in practice. To achieve equality, we would need all singular values of the Jacobian matrix $\nabla_{\mathbf{x}} f$ to be identical. However, our experiments in Section 4 show that the local dimensionality decreases rapidly with training onset; this implies that $\nabla_{\mathbf{x}} f^T \nabla_{\mathbf{x}} f$ has a non-uniform eigenspectrum (i.e., some directions being particularly elongated, corresponding to a lower overall dimension). Moreover, the volume will decrease rapidly as the smallest eigenvalue vanishes. Thus, although sharpness upper bounds the volumetric ratio and often correlates reasonably with it (see Appendix H.2), the correlation is far from perfect.

Fortunately, considering only the maximum eigenvalue instead of the product of all eigenvalues alleviates this discrepancy (recall that $\det\left(\nabla_{\mathbf{x}} f^T \nabla_{\mathbf{x}} f\right)$ in Definition 3.1 is the product of all eigenvalues). This motivates an alternative compression metric based on the largest singular value.

**Definition 3.7.** *The **Maximum Local Sensitivity (MLS)** of network $f$ is defined to be* $\mathrm{MLS}_f = \frac{1}{n} \sum_{i=1}^{n} \|\nabla_{\mathbf{x}} f(\mathbf{x}_i)\|_2$, *which is the sample mean of the largest singular value of $\nabla_{\mathbf{x}} f$.*

Intuitively, MLS is the largest possible average local change of $f(\mathbf{x})$ when the norm of the perturbation to $\mathbf{x}$ is regularized. MLS is also referred to as *adversarial robustness* or *Lipschitz constant* of the model function in Ma & Ying (2021). We now derive an analogous bound for MLS that mirrors Proposition 3.4 but avoids the product-of-eigenvalues issue.

**Proposition 3.8.** *The maximum local sensitivity is upper bounded by a sharpness-related quantity:*

$$\mathrm{MLS}_f = \frac{1}{n} \sum_{i=1}^{n} \|\nabla_{\mathbf{x}} f(\mathbf{x}_i, \boldsymbol{\theta}^*)\|_2 \leq \|\mathbf{W}\|_2 \sqrt{\frac{1}{n} \sum_{i=1}^{n} \frac{1}{\|\mathbf{x}_i\|_2^2}} S(\boldsymbol{\theta}^*)^{1/2} . \tag{11}$$

The derivation of the above bound is included in Appendix E. As an alternative measure of compressed representations, we empirically show in Appendix H.2 that MLS has a higher correlation with sharpness than the local volumetric ratio. We include more analysis of the tightness of this bound in Appendix H and discuss its connection to other works therein.

As with volumetric ratio, considering only the first layer can lead to unstable bounds dominated by prefactor variations. We therefore introduce a network-wide version that examines sensitivity at all intermediate layers, yielding bounds that more consistently predict positive correlation with sharpness across experimental settings.

**Definition 3.9.** *The **Network Maximum Local Sensitivity (NMLS)** of network $f$ is defined as the sum of* $\mathrm{MLS}_{f_l}$ *for all $l$, i.e.* $\sum_{l=1}^{L} \mathrm{MLS}_{f_l}$.

Recall that $\mathbf{x}_i^l$ is the input to the $l$-th linear/convolutional layer for sample $\mathbf{x}_i$ and $f_l$ is the mapping from the input of $l$-th layer to the final output. Extending the analysis from Proposition 3.8 to all layers yields:

**Proposition 3.10.** *The network maximum local sensitivity is upper bounded by a sharpness related quantity:*

$$\mathrm{NMLS} = \frac{1}{n} \sum_{l=1}^{L} \sum_{i=1}^{n} \|\nabla_{\mathbf{x}^l} f^l(\mathbf{x}_i^l, \boldsymbol{\theta}^*)\|_2 \leq \sqrt{\frac{1}{n} \sum_{i=1}^{n} \sum_{l=1}^{L} \frac{\|\mathbf{W}_l\|_2^2}{\|\mathbf{x}_i^l\|^2} \cdot S(\boldsymbol{\theta}^*)^{1/2}}. \tag{12}$$

The derivation is in Appendix E. The advantage of NMLS is that instead of only considering the robustness of the final output w.r.t. the input, NMLS considers the robustness of the output w.r.t. all hidden-layer representations. This allows us to derive a bound that not only considers the weights in the first linear layer but also all other linear weights. We observe in Section 4.2 that while MLS could be negatively correlated with the right-hand side of Equation (11), NMLS has a positive correlation with right-hand side of Equation (12) consistently. We do observe that MLS is still positively correlated with sharpness (Figure H.8). Therefore, the only possible reason for this negative correlation is the factor before sharpness in the MLS bound involving the first-layer weights and the quadratic mean (see Equation (11)).

**Why LVR and MLS quantify compression.** Having introduced our main compression metrics, we clarify why these quantities measure compression of neural representations. In our framework, *compression* refers to the concentration or contraction of representations when inputs are locally perturbed. LVR quantifies this via volume contraction: if a small volume in input space is mapped to a smaller volume in representation space, the network has compressed that region of its representation manifold. Mathematically, LVR $< 1$ indicates volume contraction (compression), LVR $= 1$ indicates volume preservation, and LVR $> 1$ indicates volume expansion. Similarly, MLS measures compression through sensitivity: lower MLS means the network output changes less in response to input perturbations, indicating that nearby inputs are mapped to nearby (compressed) outputs. Networks with high compression have both low LVR (contracted volumes) and low MLS (reduced sensitivity).

Importantly, our theoretical bounds do not *guarantee* that networks will compress (i.e., achieve LVR $< 1$ or small MLS). The bounds only establish that flatter minima constrain these metrics to be smaller. However, our extensive empirical results in Section 4 demonstrate that trained networks do indeed exhibit compression: we observe decreasing LVR and MLS values during training, particularly in the late stages when networks find flatter minima. The correlation between sharpness and these compression metrics validates that flatter minima are associated with more compressed representations in practice, even though the bounds themselves are upper bounds rather than guarantees of compression.

**On the role of weight norms in the bounds.** A critical aspect of our bounds is their dependence on both sharpness and weight-related prefactors (e.g., $\|\mathbf{W}\|_2/\|\mathbf{x}\|_2$ ratios). This raises a natural question: does lower sharpness *always* imply more compression? Theoretically, this is not guaranteed. Consider two networks at different interpolating solutions: if one has lower sharpness but substantially larger weight norms, the bound could be looser despite the decrease in sharpness. In principle, a decrease in sharpness might not fully counteract increases in the prefactor terms. We provide detailed empirical analysis of where tightness is lost in the chain of inequalities comprising our bounds in Appendix H.

However, despite this ambiguity in theory, the bounds may still apply usefully in practice. One important setting where this is the case is for a *fixed network architecture during training*, when the prefactors remain relatively stable as the network evolves toward flatter minima. Then weight norms (and input statistics) do not fluctuate dramatically within a training trajectory, so the sharpness term dominates the variation in the bound. This explains the strong positive correlations we observe between sharpness and compression metrics during training (e.g., Figure 2, Figure 3). In contrast, when *comparing across different network configurations*—such as in our analysis of 181 pretrained ViT models with varying architectures, datasets, and training procedures—the prefactors can vary substantially. This is precisely why we employ adaptive sharpness (which normalizes by weight norms) and normalized MLS in Figure 5: these normalized metrics account for the prefactor variations and reveal that the sharpness-compression relationship holds even across heterogeneous model families. The empirical robustness of this correlation, both within training trajectories and across diverse architectures when properly normalized, demonstrates that the bounds capture meaningful relationships even despite their theoretical limitations.

### 3.3 Local dimensionality is tied to, but not bounded by, sharpness

While LVR and MLS capture compression via volume contraction and maximum sensitivity, they do not capture the *uniformity* of compression across different directions. A network might compress representations uniformly (reducing all directions proportionally) or anisotropically (compressing certain directions more

than others). This distinction is captured by local dimensionality, which measures whether the eigenvalues of the Jacobian are uniform or concentrated in a few directions. As we will show, unlike LVR and MLS, local dimensionality cannot be directly bounded by sharpness alone, as it depends on eigenvalue ratios rather than their magnitudes.

Consider an input data point $\bar{\mathbf{x}}$ drawn from the training set: $\bar{\mathbf{x}} = \mathbf{x}_i$ for a specific $i \in \{1, \cdots, n\}$. Let the set of all possible perturbations around $\bar{\mathbf{x}}$ in the input space be samples from an isotropic normal distribution, $\mathcal{B}(\bar{\mathbf{x}})_\alpha \sim \mathcal{N}(\bar{\mathbf{x}}, \alpha\mathcal{I})$, where $C_{\mathcal{B}(\bar{\mathbf{x}})_\alpha} = \alpha\mathcal{I}$, with $\mathcal{I}$ as the identity matrix, is the covariance matrix. We first propagate $\mathcal{B}(\bar{\mathbf{x}})_\alpha$ through the network transforming each point $\mathbf{x}$ into its corresponding image $f(\mathbf{x})$. Following a Taylor expansion for points within $\mathcal{B}(\bar{\mathbf{x}})_\alpha$ in the limit $\alpha \to 0$, we have:

$$f(\mathbf{x}) = f(\bar{\mathbf{x}}) + \nabla_{\mathbf{x}}f(\bar{\mathbf{x}}, \boldsymbol{\theta}^*)^T(\mathbf{x} - \bar{\mathbf{x}}) + O(\|\mathbf{x} - \bar{\mathbf{x}}\|^2) \ . \tag{13}$$

We can express the limit of the covariance matrix $C_{f(\mathcal{B}(\mathbf{x}))}$ of the output $f(\mathbf{x})$ as

$$C_f^{\lim} := \lim_{\alpha \to 0} \frac{C_{f(\mathcal{B}(\mathbf{x})_\alpha)}}{\alpha} = \nabla_{\mathbf{x}}f(\bar{\mathbf{x}}, \boldsymbol{\theta}^*)\nabla_{\mathbf{x}}^T f(\bar{\mathbf{x}}, \boldsymbol{\theta}^*) \ . \tag{14}$$

Our covariance expressions capture the distribution of the samples in $\mathcal{B}(\bar{\mathbf{x}})_\alpha$ as they go through the network $f(\bar{\mathbf{x}}, \boldsymbol{\theta}^*)$. The local Participation Ratio based on this covariance is given by:

$$D_{\mathrm{PR}}(f(\bar{\mathbf{x}})) = \lim_{\alpha \to 0} \frac{\mathrm{Tr}[C_{f(\mathcal{B}(\mathbf{x})_\alpha)}]^2}{\mathrm{Tr}[(C_{f(\mathcal{B}(\mathbf{x})_\alpha)})^2]} = \frac{\mathrm{Tr}[C_f^{\lim}]^2}{\mathrm{Tr}[(C_f^{\lim})^2]} \tag{15}$$

(Recanatesi et al. 2022, cf. nonlocal measures in Gao et al. 2017; Litwin-Kumar et al. 2017; Mazzucato et al. 2016).

**Definition 3.11.** *The **Local Dimensionality** of a network $f$ is defined as the sample mean of local participation ratio at different input samples: $D(f) = \frac{1}{n}\sum_{i=1}^n D_{PR}(f(\mathbf{x}_i))$*

This quantity in some sense represents the sparseness of the eigenvalues of $C_f^{\lim}$: If we let $\boldsymbol{\lambda}$ be all the eigenvalues of $C_f^{\lim}$, then the local dimensionality can be written as $D_{\mathrm{PR}} = (\|\boldsymbol{\lambda}\|_1/\|\boldsymbol{\lambda}\|_2)^2$, which attains its maximum value when all eigenvalues are equal to each other, and its minimum when all eigenvalues except for the leading one are zero. Note that the quantity retains the same value when $\boldsymbol{\lambda}$ is arbitrarily scaled. As a consequence, it is hard to find a relationship between the local dimensionality and the fundamental quantity on which our bounds are based: $\|\nabla_{\mathbf{x}}f(\mathbf{x}, \boldsymbol{\theta}^*)\|_F^2$, which is $\|\boldsymbol{\lambda}\|_1$.

## 3.4 Relation to reparametrization-invariant sharpness

Dinh et al. (2017) argues that a robust sharpness metric should have the reparametrization-invariant property, meaning that scaling the neighboring linear layer weights should not change the metric. While the bounds in Proposition 3.8 and Proposition 3.10 are not strictly reparametrization-invariant, sharpness metrics that are redesigned (Tsuzuku et al., 2019) to achieve invariance can be proved to tighten our bounds (see Appendix F.1). Another more aggressive reparametrization-invariant sharpness is proposed in Andriushchenko et al. (2023); Kwon et al. (2021), and we again show that it upper-bounds input-invariant MLS in Appendix F.2. We also empirically evaluate the relative flatness (Petzka et al., 2021), which is also reparametrization-invariant in Appendix H.2, but no significant correlation is observed. Overall, we provide a novel perspective: reparametrization-invariant sharpness is characterized by the robustness of outputs to internal neural representations.

## 3.5 Connection to neural collapse and compressed neural representations

The neural collapse phenomenon (Papyan et al., 2020; Zhu et al., 2021) indicates that the within-class variance of the features in the penultimate layer vanishes at the terminal phase of training; allied studies have also found compressed neural representations at this and other points internal to trained neural networks Farrell et al. 2022 (see also Farrell et al. 2023; Shwartz-Ziv & Tishby 2017). We next show that the present sharpness-based approach can describe related properties at penultimate and intermediate network layers.

While neural collapse is often described as a *global* phenomenon (within-class variance across all samples of a class), our results are *local*, quantifying robustness of layer responses for nearby points in the input space. Nevertheless, a connection emerges when viewing within-class samples as local perturbations around their class mean, a perspective we develop below.

We first apply our method to study the penultimate-layer features. To accomplish this we can adapt the linear stability trick in Equation (5) to establish a relationship between their robustness and sharpness. More concretely, we can show that

$$\|\nabla_{\mathbf{x}} g(\mathbf{W}\mathbf{x}; \bar{\boldsymbol{\theta}})\|_F \leq \frac{\|\mathbf{W}\|_2}{\sigma_{\min}(\mathbf{W}_L)\|\mathbf{x}\|_2} \|\nabla_{\mathbf{W}} f(\mathbf{W}\mathbf{x}; \bar{\boldsymbol{\theta}})\|_F \ . \tag{16}$$

Here again, $\mathbf{W}$ denotes the first-layer weights, $\mathbf{W}_L$ denotes the last-layer linear classifier weights, and $g(x)$ is the penultimate-layer feature. $\sigma_{\min}(\mathbf{W}_L)$ is defined as the square root of the smallest eigenvalue of $\mathbf{W}_L^T \mathbf{W}_L$. The proof is given in Appendix A.

To extend this analysis to representations before the penultimate layer, note that inequality (16) extends to any middle-layer representation, as detailed in Appendix A, so that very similar conclusions apply directly.

Let the feature dimension be $d$ and the number of classes be $K$. Then, $\mathbf{W}_L \in \mathbb{R}^{K \times d}$, and $\sigma_{\min}(\mathbf{W}_L) = 0$ if $d > K$; otherwise, it is the smallest singular value of $\mathbf{W}_L$. An effective bound on the robustness of penultimate-layer (or other internal-layer) features is obtained when $d \leq K$, i.e., when the number of classes is at least as large as the feature dimension. This condition is naturally satisfied in settings such as language modeling, retrieval systems, and face recognition applications, where the number of classes (vocabulary size, number of items, number of identities) is typically much larger than the feature dimension.

Interestingly, these are also settings where generalized neural collapse phenomena have been observed (Jiang et al., 2023). While our bounds are local (quantifying robustness to small input perturbations), a potential connection to neural collapse emerges when we consider samples from the same class as local perturbations around the class mean. In neural collapse, the within-class variance of features vanishes, meaning all samples from a class produce nearly identical features. This can be viewed through our framework as extreme robustness: if the network maps all within-class samples (which differ from the class mean by small to moderate perturbations) to the same feature representation, the features exhibit strong compression and robustness properties that our bounds capture locally. This said, the relationship between our local robustness bounds and the global geometric structure of neural collapse, including other aspects not considered here (e.g., the simplex equiangular tight frame geometry) remains an open question for future investigation.

## 4 Experiments

*All networks are trained with MSE (quadratic) loss except for the pretrained ViTs in Section 4.3.*

### 4.1 Sharpness and compression metrics during training: verifying the theory

The theoretical results derived above show that when the training loss is low, measures of compression of the network's representation are upper-bounded by a function of the sharpness of the loss function in parameter space. This links sharpness and representation compression: the flatter the loss landscape, the lower the upper bound on the representation's compression metrics.

To empirically verify whether these bounds are sufficiently tight to show a clear relationship between sharpness and representation compression, we trained a VGG-11 network (Simonyan & Zisserman, 2015) to classify images from the CIFAR-10 dataset (Krizhevsky, 2009) and calculated the (approximate) sharpness (Equation (3)), the log volumetric ratio (Equation (7)), MLS (Definition 3.7) and NMLS (Definition 3.9) during the training phase (Figure 2 and Figure 3).

We trained the network using SGD on CIFAR-10 images and explored the influence of two specific parameters that previous work has shown to affect the network's sharpness: learning rate and batch size (Jastrzebski

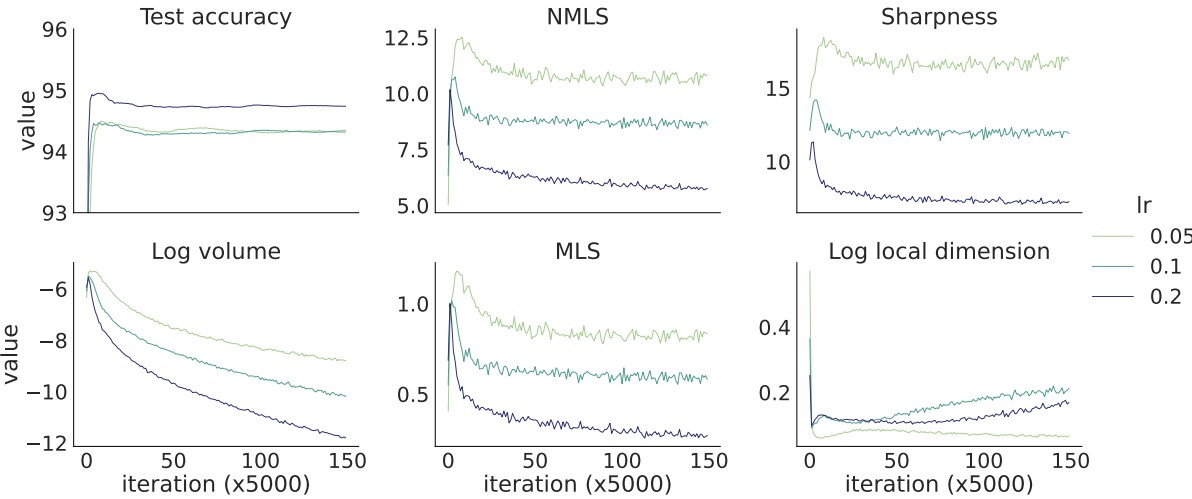

Figure 2: Trends in key variables across SGD training of the VGG-11 network with fixed batch size (equal to 20) and varying learning rates (0.05, 0.1 and 0.2). Higher learning rates lead to lower sharpness and hence stronger compression. From left to right: Test accuracy, NMLS, sharpness (square root of Equation (3)), log volumetric ratio (Equation (7)), MLS, and local dimensionality of the network output (Equation (15)).

et al., 2018). For each combination of learning rate and batch size parameters, we computed all quantities across 100 input samples and averaged across five different random initializations for network weights.

In Figure 2, we study the link between sharpness and representation compression with a fixed batch size (of 20). We observe that when the network reaches the interpolation regime, that is, when training loss is extremely low, the sharpness decreases with compression metrics, including MLS, NMLS, and volume. The trend is consistent across multiple learning rates for a fixed batch size, and MLS and NMLS match better with the trend of sharpness.

In Figure 3, we repeated the experiments while keeping the learning rate fixed at 0.1 and varying the batch size. The same broadly consistent trends emerged, linking a decrease in the sharpness to a compression in the neural representation. However, we also found that while sharpness stops decreasing after about $5 \cdot 10^4$ iterations for a batch size of 32, the volume continues to decrease as learning proceeds. This suggests that other mechanisms, beyond sharpness, may be at play in driving the compression of volumes.

We repeated the experiments with an MLP trained on the FashionMNIST dataset (Xiao et al., 2017) (Figure I.10 and Figure I.11). The sharpness again follows the same trend as MLS and NMLS, consistent with our bound. The volume continues to decrease after the sharpness plateaus, albeit at a much slower rate, again matching our theory, while suggesting that an additional factor may be involved in its decrease.

### 4.2 Correlation between compression metrics and their sharpness-related bounds

We next test the correlation between both sides of the bounds that we derive more generally. In Figure 4, we show pairwise scatter plots between MLS (resp. NMLS) and the sharpness-related bound on MLS (resp. NMLS) (we include an exhaustive set of correlation matrices in Appendix H.2). Interestingly, we find that quantities that only consider a single layer of weights, such as the relative flatness (Petzka et al., 2021) and the bound on the MLS (Proposition 3.8), can exhibit a negative correlation between both sides of the bound in some cases (Figures 4, H.7 and H.8). This demonstrates the necessity of introducing NMLS to explain the relationship between sharpness and compression. Nevertheless, we find that all compression metrics, such as MLS, NMLS, and LVR, introduced in Section 3, correlate well with sharpness in all of our experiments

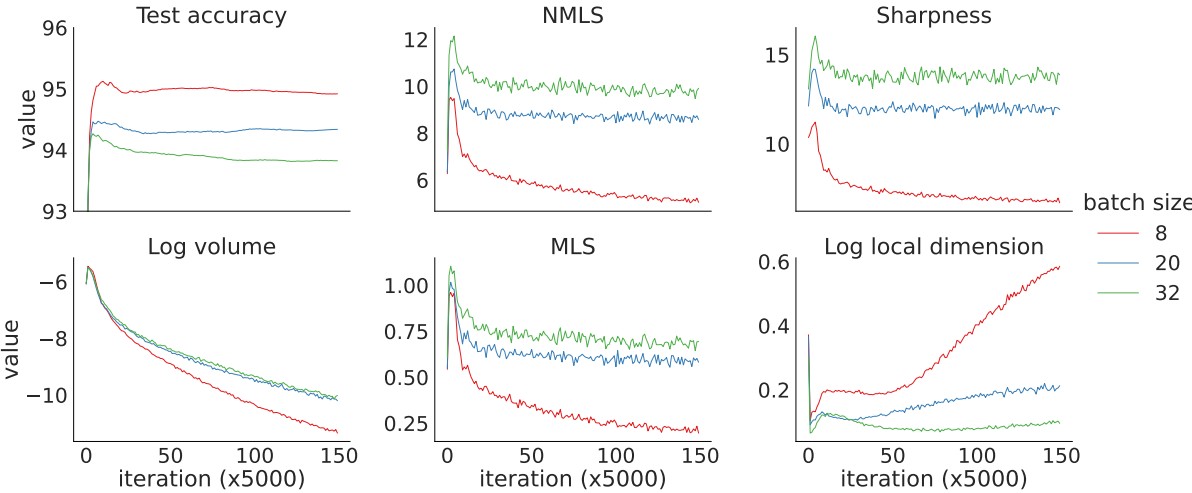

Figure 3: Trends in key variables across SGD training of the VGG-11 network with fixed learning rate size (equal to 0.1) and varying batch size (8, 20, and 32). Smaller batch sizes lead to lower sharpness and hence stronger compression. From left to right in row-wise order: test accuracy, NMLS, sharpness (square root of Equation (3)), log volumetric ratio (Equation (7)), MLS, and local dimensionality of the network output (Equation (15)).

(Appendix H.2). Although the bound in Proposition 3.4 is loose, the log LVR still correlates positively with sharpness and MLS.

### 4.3 Empirical evidence in Vision Transformers (ViTs)

Since our theory applies to linear and convolutional layers as well as residual layers (Appendix B), relationships among sharpness and compression, as demonstrated above for VGG-11 and MLP networks, it should hold more generally in modern architectures such as the Vision Transformer (ViT) and its variants. However, naive ways of evaluating the quantities discussed in previous sections are computationally prohibitive. Instead, we look at the MLS normalized by the norm of the input and the elementwise-adaptive sharpness defined in Andriushchenko et al. (2023); Kwon et al. (2021). Both of the metrics can be estimated efficiently for large networks. Specifically, in Figure 5 we plot the normalized MLS against the elementwise-adaptive sharpness. The analytical relationship between the normalized MLS and the elementwise-adaptive sharpness and the details of the numerical approximation we used are given in Appendix F.2 and Appendix G respectively. For all the models, we attach a sigmoid layer to the output logits and use MSE loss to calculate the adaptive sharpness. Figure 5 shows the results for 181 pretrained ViT models provided by the `timm` package (Wightman, 2019). We observe that there is a general trend that lower sharpness indeed implies lower MLS. However, there are also outlier clusters, with data corresponding to the same model class; an interesting future direction would be to understand the mechanisms driving this outlier behavior. Interestingly, we did not observe this correlation between unnormalized metrics, indicating that weight scales should be taken into account when comparing between different models.

### 4.4 Sharpness and local dimensionality

A priori, it is unclear whether the local dimensionality of the neural representations should increase or decrease as the volume is compressed. For example, the volume could decrease while maintaining its overall form and symmetry, thus preserving its dimensionality. Alternatively, one or more of the directions in the relevant tangent space could be selectively compressed, leading to an overall reduction in dimensionality.

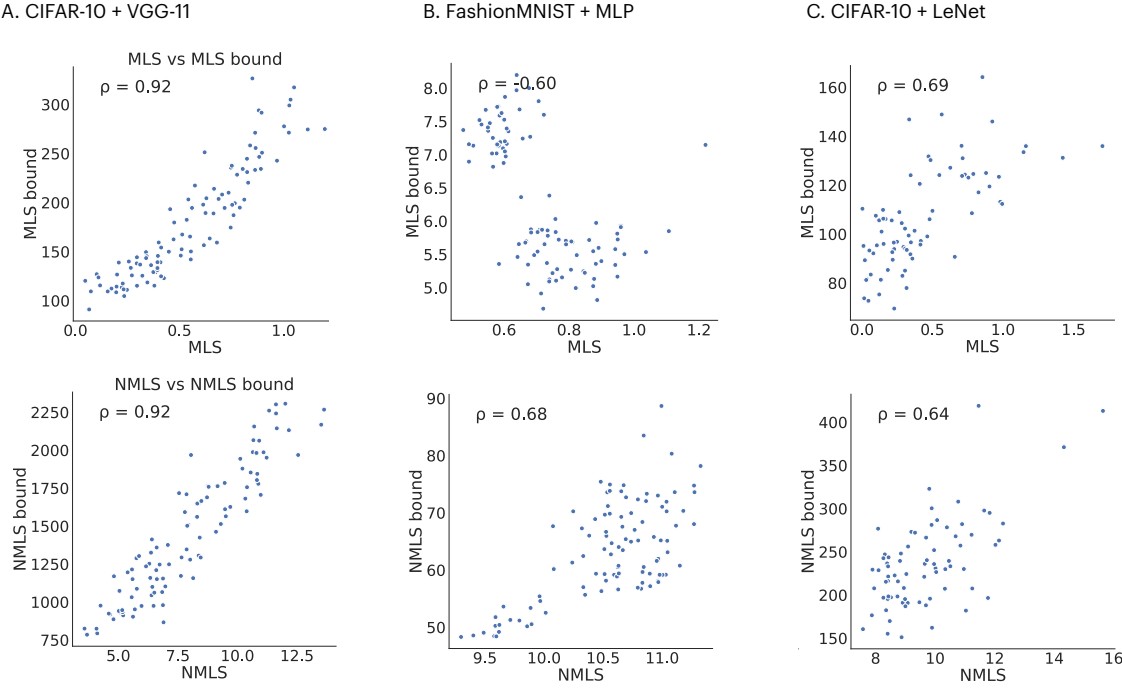

Figure 4: We trained 100 different models for each combination of datasets and networks by varying learning rates, batch size, and random initializations. Pairwise scatter plots between MLS (resp. NMLS) and the sharpness-related bound on MLS (resp. NMLS) are shown here. For MLS (resp. NMLS) bound see Proposition 3.8 (resp. Proposition 3.10). The Pearson correlation coefficient $\rho$ is shown in the top-left corner for each scatter plot. See Appendix H.2 for the full pairwise scatter matrix.

Figure 2 and Figure 3 show our experiments computing the local dimensionality over the course of training. Here, we find that the local dimensionality of the representation decreases as the loss decreases to near 0, which is consistent with the viewpoint that the network compresses representations in feature space as much as possible, retaining only the directions that code for task-relevant features (Berner et al., 2020; Cohen et al., 2020). However, the local dimensionality exhibits unpredictable behavior that cannot be explained by the sharpness once the network finds an approximate interpolation solution and training continues. Further experiments also demonstrate a weaker correlation of sharpness and local dimensionality compared to other metrics such as MLS and volume (Figures H.7 to H.9). This discrepancy is consistent with the bounds established by our theory, which only bound the numerator of Equation (15). It is also consistent with the property of local dimensionality that we described in Section 3.3 overall: it encodes the sparseness of the eigenvalues but it does not encode the magnitude of them. This shows how local dimensionality is a distinct quality of network representations compared with volume, and is driven by mechanisms that differ from sharpness alone. We emphasize that the dimensionality we study here is a local measure, on the finest scale around a point on the "global" manifold of unit activities; dimension on larger scales (i.e., across categories or large sets of task inputs (Farrell et al., 2022; Kothapalli et al., 2022; Zhu et al., 2021; Ansuini et al., 2019; Recanatesi et al., 2019; Papyan et al., 2020)) may show different trends.

# 5 Discussion: connection to generalization

So far we have avoided remarking on implications of our results for the relationship between generalization and sharpness or compression because generalization is *not* the main focus of our work. However, our work may have implications for future research on this relationship, and we briefly discuss this here. Our theoretical

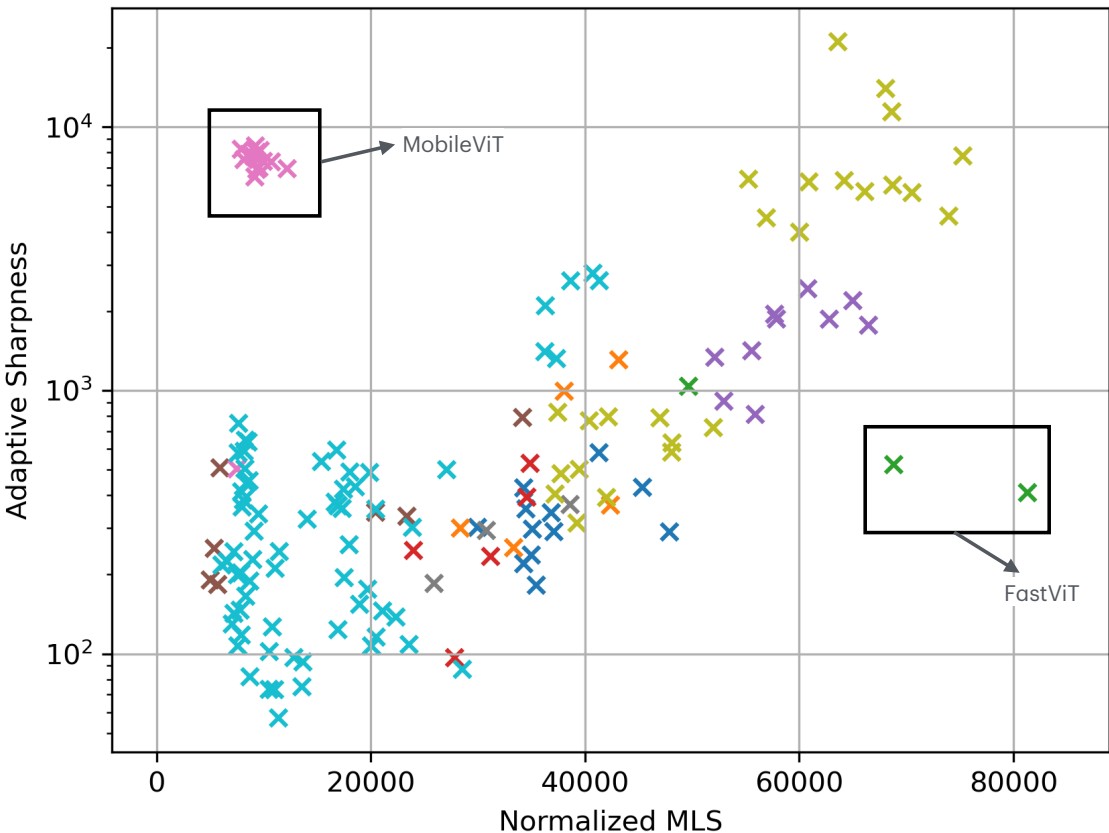

Figure 5: Adaptive sharpness vs Normalized MLS for 181 ViT models and variants. Different colors represent different model classes. For most models, there is a positive correlation between Sharpness and MLS. However, outlier clusters also exist, for MobileViT (Mehta & Rastegari, 2022) models in the upper left corner, and two FastViT (Vasu et al., 2023) models in the lower right corner.

results establish that sharpness directly controls compression of neural representations—specifically, how robust intermediate representations are to input perturbations. By "robustness of representations," we mean that small changes to the input produce proportionally small changes in the network's internal feature representations (quantified by low LVR and MLS). This compression property is desirable in adversarial settings where we want representations to remain stable under input perturbations.

However, representation robustness alone does not guarantee better generalization. Ma & Ying (2021) gave a generalization bound based on adversarial robustness, but the theorem requires that most test data lie close to the training data, which demands extensive training coverage. In contrast, our correlation analysis in Appendix H.2 shows that sharpness consistently has a higher correlation with compression metrics than with the generalization gap. Additional experiments in Appendix I show that lower sharpness does not always imply better test accuracy.

This phenomenon is also observed in applied machine learning literature. For example, it is well-established that large learning rates lead to lower sharpness (flatter minima), a relationship we also observe in our Figure 2 and consistent with Cohen et al. (2020) and Wu et al. (2022). If low sharpness directly implied better generalization, we would expect large learning rates to improve out-of-distribution (OOD) performance. However, Wortsman et al. (2022) demonstrates the opposite: large learning rates can severely hurt OOD generalization. This shows that low sharpness alone—and the compressed representations it produces—does not guarantee superior generalization.

Thus, our work offers a possible resolution to the contradictory results on the relationship between (reparametrization-invariant) sharpness and generalization (Wen et al., 2023; Andriushchenko et al., 2023): sharpness directly determines representation compression, but whether this compression translates to superior generalization may depend on other latent factors beyond just sharpness.

One possible factor is that compression metrics (LVR, MLS, NMLS) measure robustness to *small random perturbations*—networks with low sharpness have representations that are insensitive to noise. However, good generalization sometimes requires networks to remain sensitive to *semantically meaningful differences* in the input. To illustrate this distinction, consider a large language model (LLM) performing a "needle in a haystack" test, where it must extract specific information from a long context. When the query changes meaningfully (e.g., asking for a different fact), the output should change substantially to provide the correct answer—the network must be sensitive to the semantic content of the query. Yet simultaneously, the network should be robust to small random perturbations in the text, such as typos or minor paraphrasing. Sharpness controls the latter (robustness to random noise), but whether this robustness aids generalization depends on the nature of the distribution shift at test time: if test inputs primarily differ from training via noise-like perturbations, compression helps; if they differ in task-relevant semantic features that require discrimination, excessive compression may reduce the network's ability to distinguish these meaningful differences.

## 6  Summary and Conclusion

This work presents a dual perspective, uniting views in both parameter and feature space, of several properties of trained neural networks. We identify two representation compression metrics that are bounded by sharpness – local volumetric ratio and maximum local sensitivity – and give new explicit formulas for these bounds. We conducted extensive experiments with feedforward, convolutional, and attention-based networks and found that the predictions of these bounds are borne out for these networks, illustrating how MLS in particular is strongly correlated with sharpness. Overall, we establish explicit links between sharpness properties in parameter spaces and compression and robustness properties in the feature space.

By demonstrating both how these links can be tight, and how and when they may also become loose, we propose that taking this dual perspective can bring more clarity to the often confusing question of what sharpness actually quantifies in practice. Indeed, many works, as reviewed in the introduction, have demonstrated how sharpness can lead to generalization, but recent studies have established contradictory results. Therefore, our work contributes to the further exploration of sharpness-based methods for improving neural network performance. Future directions include tightening the derived bounds given knowledge of the data distribution, and investigating the conditions under which representation compression translates to improved generalization.

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

# A   Linear stability trick on penultimate-layer and middle-layer features

We define $g(x)$ to be the penultimate-layer features such that $f(x) = \mathbf{W}_L g(\mathbf{W}x) + b$. With slight abuse of notation, we define $J = \frac{\partial g(\mathbf{Wx})}{\partial \mathbf{Wx}}$. Similar to Equation (4), we have

$$
\begin{aligned}
\|\nabla_{\mathbf{W}} f(\mathbf{Wx}; \bar{\boldsymbol{\theta}})\|_F &= \|\mathbf{W}_L J\|_F \|\mathbf{x}\|_2 \\
\nabla_{\mathbf{x}} g(\mathbf{Wx}; \bar{\boldsymbol{\theta}}) &= J\mathbf{W} \ ,
\end{aligned}
\tag{17}
$$

**Lemma A.1.** $\|AB\|_F^2 \geq \lambda_{\min}(A^T A)\|B\|_F^2$, where $\lambda_{\min}$ is the smallest eigenvalue.

*Proof.* By the definition of Frobenius norm,

$$
\|AB\|_F^2 = \operatorname{Tr}(ABB^T A^T) = \operatorname{Tr}(A^T ABB^T).
\tag{18}
$$

From Fang et al. (1994), we have that for positive semidefinite matrices $P$ and $Q$,

$$
\lambda_{\min}(P)\operatorname{Tr}(Q) \leq \operatorname{Tr}(PQ)
\tag{19}
$$

Therefore,

$$
\operatorname{Tr}(A^T ABB^T) \geq \lambda_{\min}(A^T A)\operatorname{Tr}(BB^T) = \lambda_{\min}(A^T A)\|B\|_F^2
\tag{20}
$$

$\square$

As a result, $\|\mathbf{W}_L J\|_F \geq \sqrt{\lambda_{\min}(\mathbf{W}_L^T \mathbf{W}_L)}\|J\|_F$. Let $d$ be the feature dimension, and $K$ be the number of classes, and $\mathbf{W}_L \in \mathbb{R}^{K \times d}$. Then, $\lambda_{\min}(\mathbf{W}_L^T \mathbf{W}_L)$ vanishes when $d > K$, otherwise $\sqrt{\lambda_{\min}(\mathbf{W}_L^T \mathbf{W}_L)} = \sigma_{\min}(\mathbf{W}_L)$, the smallest singular value of $\mathbf{W}_L$. Therefore,

$$
\begin{aligned}
\|\nabla_{\mathbf{x}} g(\mathbf{Wx}; \bar{\boldsymbol{\theta}})\|_F &= \|J\mathbf{W}\|_F \\
&\leq \|J\|_F \|\mathbf{W}\|_2 \\
&\leq \frac{\|\mathbf{W}_L J\|_F}{\sqrt{\lambda_{\min}(\mathbf{W}_L^T \mathbf{W}_L)}}\|\mathbf{W}\|_2 \\
&= \frac{\|\nabla_{\mathbf{W}} f(\mathbf{Wx}; \bar{\boldsymbol{\theta}})\|_F}{\|\mathbf{x}\|_2 \sqrt{\lambda_{\min}(\mathbf{W}_L^T \mathbf{W}_L)}}\|\mathbf{W}\|_2
\end{aligned}
\tag{21}
$$

More generally, if we consider arbitrary intermediate-layer representations, we write $f(x) = h \circ g(\mathbf{W}x)$, where $g$ is the transformation from linear transformed input to the intermediate-layer representations and $h$ is the mapping from the representations to the output of the network. Then Equation (17) becomes

$$
\begin{aligned}
\|\nabla_{\mathbf{W}} f(\mathbf{Wx}; \bar{\boldsymbol{\theta}})\|_F &= \|J_h J_g\|_F \|\mathbf{x}\|_2 \\
\nabla_{\mathbf{x}} g(\mathbf{Wx}; \bar{\boldsymbol{\theta}}) &= J\mathbf{W} \ ,
\end{aligned}
\tag{22}
$$

where $J_g = \frac{\partial g(\mathbf{Wx})}{\partial \mathbf{Wx}}$ and $J_h = \frac{\partial f(\mathbf{Wx})}{\partial g(\mathbf{Wx})}$. Therefore, similar to Equation (21), we have

$$
\|\nabla_{\mathbf{x}} g(\mathbf{Wx}; \bar{\boldsymbol{\theta}})\|_F = \frac{\|\nabla_{\mathbf{W}} f(\mathbf{Wx}; \bar{\boldsymbol{\theta}})\|_F}{\|\mathbf{x}\|_2 \sqrt{\lambda_{\min}(J_h^T J_h)}}\|\mathbf{W}\|_2
\tag{23}
$$

## B    Adaptation of Inequality 6 to Residual Layers

We need to slightly adapt the proof in Equations (4) and (5). Consider a network whose first layer has a residual connection: $y = g(x + f(Wx))$, where $f$ is the nonlinearity with bias (e.g. $f(x) = \tanh(x + b)$), and $g$ is the rest of the mappings in the network. Then we have

$$\|\nabla_W g(x + f(Wx))\|_F = \|JK\|_F \|x\|_2$$
$$\nabla_x g(x + f(Wx)) = J + JKW \tag{24}$$

where $J = \frac{\partial g(x + f(Wx))}{\partial(x + f(Wx))}$ and $K = \frac{\partial f(Wx)}{\partial(Wx)}$.

Therefore, $\|\nabla_x g(x + f(Wx))\|_2 \leq \|J\|_2 + \|JK\|_2 \|W\|_2 \leq \|J\|_2 + \frac{\|\nabla_W g(x + f(Wx))\|_F}{\|x\|_2} \|W\|_2$. Now, we get the bound for the *difference* between MLS of input and the MLS of input to the next layer:

$$\|\nabla_x g(x + f(Wx))\|_2 - \|J\|_2 \leq \frac{\|\nabla_W g(x + f(Wx))\|_F}{\|x\|_2} \|W\|_2 \tag{25}$$

Notice that if we apply this inequality to every residual layer in the network, and sum the left-hand side, we will get a telescoping sum on the left-hand side. Assuming the last layer is linear with weights $W_L$, we get $\|\nabla_x g(x + f(W_1 x))\|_2 - \|W_L\|_2 \leq \sum_{l=1}^{L-1} \frac{\|W_l\|_2}{\|x_l\|_2} \|\nabla_W g_l(x_l + f(W_l x_l))\|_F$. The right-hand side is bounded by sharpness due to Cauchy, see also Equation (42).

## C    Proof of Equation (3)

**Lemma C.1.** *If $\boldsymbol{\theta}$ is an approximate interpolation solution, i.e. $\|f(\mathbf{x}_i, \boldsymbol{\theta}) - \mathbf{y}_i\| < \varepsilon$ for $i \in \{1, 2, \cdots, n\}$, and second derivatives of the network function $\|\nabla^2_{\theta_j} f(\mathbf{x}_i, \boldsymbol{\theta})\| < M$ is bounded, then*

$$S(\boldsymbol{\theta}^*) = \frac{1}{n} \sum_{i=1}^n \|\nabla_{\boldsymbol{\theta}} f(\mathbf{x}_i, \boldsymbol{\theta}^*)\|_F^2 + O(\varepsilon) \tag{26}$$

*Proof.* Using basic calculus we get

$$S(\boldsymbol{\theta}) = \mathrm{Tr}(\nabla^2 L(\boldsymbol{\theta}))$$
$$= \frac{1}{2n} \sum_{i=1}^n \mathrm{Tr}(\nabla^2_{\boldsymbol{\theta}} \|f(\mathbf{x}_i, \boldsymbol{\theta}) - \mathbf{y}_i\|^2)$$
$$= \frac{1}{2n} \sum_{i=1}^n \mathrm{Tr}\,\nabla_{\boldsymbol{\theta}}(2(f(\mathbf{x}_i, \boldsymbol{\theta}) - \mathbf{y}_i)^T \nabla_{\boldsymbol{\theta}} f(\mathbf{x}_i, \boldsymbol{\theta}))$$
$$= \frac{1}{n} \sum_{i=1}^n \sum_{j=1}^m \frac{\partial}{\partial \boldsymbol{\theta}_j}((f(\mathbf{x}_i, \boldsymbol{\theta}) - \mathbf{y}_i)^T \nabla_{\boldsymbol{\theta}} f(\mathbf{x}_i, \boldsymbol{\theta}))_j$$
$$= \frac{1}{n} \sum_{i=1}^n \sum_{j=1}^m \frac{\partial}{\partial \boldsymbol{\theta}_j}(f(\mathbf{x}_i, \boldsymbol{\theta}) - \mathbf{y}_i)^T \nabla_{\boldsymbol{\theta}_j} f(\mathbf{x}_i, \boldsymbol{\theta})$$
$$= \frac{1}{n} \sum_{i=1}^n \sum_{j=1}^m \|\nabla_{\boldsymbol{\theta}_j} f(\mathbf{x}_i, \boldsymbol{\theta})\|_2^2 + (f(\mathbf{x}_i, \boldsymbol{\theta}) - \mathbf{y}_i)^T \nabla^2_{\boldsymbol{\theta}_j} f(\mathbf{x}_i, \boldsymbol{\theta})$$
$$= \frac{1}{n} \sum_{i=1}^n \|\nabla_{\boldsymbol{\theta}} f(\mathbf{x}_i, \boldsymbol{\theta})\|_F^2 + \frac{1}{n} \sum_{i=1}^n \sum_{j=1}^m (f(\mathbf{x}_i, \boldsymbol{\theta}) - \mathbf{y}_i)^T \nabla^2_{\boldsymbol{\theta}_j} f(\mathbf{x}_i, \boldsymbol{\theta}).$$

Therefore

$$\left| S(\boldsymbol{\theta}) - \frac{1}{n}\sum_{i=1}^{n}\|\nabla_{\boldsymbol{\theta}}f(\mathbf{x}_i,\boldsymbol{\theta})\|_F^2 \right| < \frac{1}{n}\sum_{i=1}^{n}\sum_{j=1}^{m}|(f(\mathbf{x}_i,\boldsymbol{\theta})-\mathbf{y}_i)^T\nabla_{\boldsymbol{\theta}_j}^2 f(\mathbf{x}_i,\boldsymbol{\theta})| < mM\varepsilon = O(\varepsilon). \qquad (27)$$

$\square$

In other words, when the network reaches zero training error and enters the interpolation phase (i.e., it fits all training data correctly), Equation (3) will be a good enough approximation of the sharpness because the quadratic training loss is sufficiently small.

## D    Proof of Equation (8), Proposition 3.4 and Proposition 3.6

### D.1    Proof of Equation (8)

We prove the bound on the local volumetric ratio using the arithmetic-geometric mean inequality. Let $J = \nabla_{\mathbf{x}}f \in \mathbb{R}^{N \times M}$ with singular values $\sigma_1, \ldots, \sigma_N > 0$. By the AM-GM inequality applied to the squared singular values:

$$
\begin{aligned}
\sqrt{\det(JJ^T)} = \sqrt{\prod_{i=1}^{N}\sigma_i^2} &= \prod_{i=1}^{N}\sigma_i \\
&\leq \left(\frac{1}{N}\sum_{i=1}^{N}\sigma_i^2\right)^{N/2} \quad \text{(AM-GM inequality)} \\
&= \left(\frac{\operatorname{Tr}(JJ^T)}{N}\right)^{N/2} \quad \text{(eigenvalues of } JJ^T \text{ are } \sigma_i^2\text{)} \\
&= \left(\frac{\operatorname{Tr}(J^TJ)}{N}\right)^{N/2} \quad \text{(trace identity: } \operatorname{Tr}(AB)=\operatorname{Tr}(BA)\text{)} \\
&= \left(\frac{\|J\|_F^2}{N}\right)^{N/2} \quad \text{(Frobenius norm: } \|J\|_F^2 = \operatorname{Tr}(J^TJ)\text{)}.
\end{aligned}
\qquad (28)
$$

This completes the proof of Equation (8).

### D.2    Proof of Proposition 3.4 and Proposition 3.6

For notational simplicity, we write $f_i := f(\mathbf{x}_i, \boldsymbol{\theta}^*)$ in what follows. Because of Equation (5), we have the following inequality due to Cauchy-Swartz inequality,

$$
\begin{aligned}
\frac{1}{n}\sum_{i=1}^{n}\|\nabla_{\mathbf{x}}f_i\|_F^k &\leq \|\mathbf{W}\|_2^k \frac{1}{n}\sum_{i=1}^{n}\frac{\|\nabla_{\mathbf{W}}f_i\|_F^k}{\|\mathbf{x}_i\|_2^k} \\
&\leq \frac{1}{n}\|\mathbf{W}\|_2^k\sqrt{\sum_{i=1}^{n}\frac{1}{\|\mathbf{x}_i\|_2^{2k}}} \cdot \sqrt{\sum_{i=1}^{n}\|\nabla_{\mathbf{W}}f_i\|_F^{2k}}.
\end{aligned}
\qquad (29)
$$

Since the input weights $\mathbf{W}$ is just a part of all the weights $(\boldsymbol{\theta})$ of the network, we have $\|\nabla_{\mathbf{W}}f_i\|_F^k \leq \|\nabla_{\boldsymbol{\theta}}f_i\|_F^k$.

We next show the correctness of Proposition 3.4 with a standard lemma.

**Lemma D.1.** *For vector* $\mathbf{x}$, $\|\mathbf{x}\|_p \geq \|\mathbf{x}\|_q$ *for* $1 \leq p \leq q \leq \infty$.

*Proof.* First we show that for $0 < k < 1$, we have $(|a|+|b|)^k \leq |a|^k + |b|^k$. It's trivial when either $a$ or $b$ is 0. So W.L.O.G, we can assume that $|a| < |b|$, and divide both sides by $|b|^k$. Therefore it suffices to show that

for $0 < t < 1$, $(1+t)^k < t^k + 1$. Let $f(t) = (1+t)^k - t^k - 1$, then $f(0) = 0$, and $f'(t) = k(1+t)^{k-1} - kt^{k-1}$. Because $k - 1 < 0$, $1 + t > 1$ and $t < 1$, $t^{k-1} > 1 > (1+t)^{k-1}$. Therefore $f'(t) < 0$ and $f(t) < 0$ for $0 < t < 1$. Combining all cases, we have $(|a| + |b|)^k \leq |a|^k + |b|^k$ for $0 < k < 1$. By induction, we have $(\sum_n |a_n|)^k \leq \sum_n |a_n|^k$.

Now we can prove the lemma using the conclusion above,

$$\left(\sum_n |x_n|^q\right)^{1/q} = \left(\sum_n |x_n|^q\right)^{p/q \cdot 1/p} \leq \left(\sum_n (|x_n|^q)^{p/q}\right)^{1/p} = \left(\sum_n |x_n|^p\right)^{1/p}$$

$\square$

Now we can prove Proposition 3.4

**Proposition.** *The local volumetric ratio is upper bounded by a sharpness related quantity:*

$$dV_{f(\boldsymbol{\theta}^*)} \leq \frac{N^{-N/2}}{n} \sum_{i=1}^n \|\nabla_{\mathbf{x}} f(\mathbf{x}_i, \boldsymbol{\theta}^*)\|_F^N \leq \frac{1}{n}\sqrt{\sum_{i=1}^n \frac{\|\mathbf{W}\|_2^{2N}}{\|\mathbf{x}_i\|_2^{2N}} \left(\frac{nS(\boldsymbol{\theta}^*)}{N}\right)^{N/2}} \tag{30}$$

*for all $N \geq 1$.*

*Proof.* Take the $x_i$ in Lemma D.1 to be $\|\nabla_{\boldsymbol{\theta}} f(\mathbf{x}_i, \boldsymbol{\theta}^*)\|_F^2$ and let $p = 1, q = k$, then we get

$$\left(\sum_{i=1}^n (\|\nabla_{\boldsymbol{\theta}} f_i\|_F^2)^k\right)^{1/k} \leq \sum_{i=1}^n \|\nabla_{\boldsymbol{\theta}} f_i\|_F^2. \tag{31}$$

Therefore,

$$\frac{1}{n}\|\mathbf{W}\|_2^k \sqrt{\sum_{i=1}^n \frac{1}{\|\mathbf{x}_i\|_2^{2k}}} \cdot \sqrt{\sum_{i=1}^n \|\nabla_{\mathbf{W}} f_i\|_F^{2k}} \leq n^{k/2-1}\|\mathbf{W}\|_2^k \sqrt{\sum_{i=1}^n \frac{1}{\|\mathbf{x}_i\|_2^{2k}}} \left(\frac{1}{n}\sum_{i=1}^n \|\nabla_{\boldsymbol{\theta}} f_i\|_F^2\right)^{k/2}$$

$$= n^{k/2-1}\|\mathbf{W}\|_2^k \sqrt{\sum_{i=1}^n \frac{1}{\|\mathbf{x}_i\|_2^{2k}}} S(\boldsymbol{\theta}^*)^{k/2} \tag{32}$$

$\square$

Next, we show that the first inequality in Equation (32) can be tightened by considering all linear layer weights.

**Proposition.** *The network volumetric ratio is upper bounded by a sharpness related quantity:*

$$\sum_{l=1}^L dV_{f_l} \leq \frac{N^{-N/2}}{n} \sum_{l=1}^L \sum_{i=1}^n \|\nabla_{\mathbf{x}^l} f_i^l\|_F^N \leq \frac{1}{n}\sqrt{\sum_{l=1}^L \sum_{i=1}^n \frac{\|\mathbf{W}_l\|_2^{2N}}{\|\mathbf{x}_i^l\|_2^{2N}} \cdot \left(\frac{nS(\boldsymbol{\theta}^*)}{N}\right)^{N/2}}. \tag{33}$$

*Proof.* Recall that the input to the $l$-th linear layer is $x_i^l$ for $l = 1, 2, \cdots, L$. In particular, $x_i^1$ is the input of the entire network. Similarly, $\mathbf{W}_l$ is the weight matrix of $l$-th linear/convolutional layer. With a slight abuse of notation, we use $f^l$ to denote the mapping from the activity of $l$-th layer to the final output, and $f_i^l := f^l(\mathbf{x}_i, \boldsymbol{\theta}^*)$. We can apply Cauchy-Swartz inequality again to get

$$\frac{1}{n}\sum_{l=1}^L \sum_{i=1}^n \|\nabla_{\mathbf{x}^l} f_i^l\|_F^k \leq \frac{1}{n}\sum_{l=1}^L \sqrt{\sum_{i=1}^n \frac{\|\mathbf{W}_l\|_2^{2k}}{\|\mathbf{x}_i^l\|_2^{2k}}} \cdot \sqrt{\sum_{i=1}^n \|\nabla_{\mathbf{W}_l} f_i^l\|_F^{2k}}$$

$$\leq \sqrt{\frac{1}{n}\sum_{l=1}^L \sum_{i=1}^n \frac{\|\mathbf{W}_l\|_2^{2k}}{\|\mathbf{x}_i^l\|_2^{2k}}} \cdot \sqrt{\frac{1}{n}\sum_{l=1}^L \sum_{i=1}^n \|\nabla_{\mathbf{W}_l} f_i^l\|_F^{2k}}. \tag{34}$$

Using Lemma D.1 again we have

$$
\left(\sum_{l=1}^{L}(\|\nabla_{\boldsymbol{W}_l}f_i^l\|_F^2)^k\right)^{1/k} \leq \sum_{l=1}^{L}\|\nabla_{\boldsymbol{W}_l}f_i^l\|_F^2 = \|\nabla_{\boldsymbol{\theta}}f_i\|_F^2,
$$

$$
\left(\sum_{i=1}^{n}(\|\nabla_{\boldsymbol{\theta}}f_i\|_F^2)^k\right)^{1/k} \leq \sum_{i=1}^{n}\|\nabla_{\boldsymbol{\theta}}f_i\|_F^2 = nS(\boldsymbol{\theta}^*),
$$

(35)

The second equality holds because both sides represent the same gradients in the computation graph. Therefore from Equation (34), we have

$$
\frac{1}{n}\sum_{l=1}^{L}\sum_{i=1}^{n}\|\nabla_{\mathbf{x}^l}f_i^l\|_F^k \leq \sqrt{\frac{1}{n}\sum_{l=1}^{L}\sum_{i=1}^{n}\frac{\|\mathbf{W}_l\|_2^{2k}}{\|\mathbf{x}_i^l\|_2^{2k}}} \cdot \sqrt{n^{k-1}S(\boldsymbol{\theta}^*)^k}
$$

(36)

$\square$

## E  Proof of Proposition 3.8 and Proposition 3.10

Below we give the proof of Proposition 3.8.

**Proposition.** *The maximum local sensitivity is upper bounded by a sharpness related quantity:*

$$
\mathrm{MLS} = \frac{1}{n}\sum_{i=1}^{n}\|\nabla_{\mathbf{x}}f(\mathbf{x}_i,\boldsymbol{\theta}^*)\|_2 \leq \|\mathbf{W}\|_2\sqrt{\frac{1}{n}\sum_{i=1}^{n}\frac{1}{\|\mathbf{x}_i\|_2^2}}S(\boldsymbol{\theta}^*)^{1/2} .
$$

(37)

*Proof.* From Equation (5), we get

$$
\mathrm{MLS} = \frac{1}{n}\sum_{i=1}^{n}\|\nabla_{\mathbf{x}}f_i\|_2 \leq \|\mathbf{W}\|_2\frac{1}{n}\sum_{i=1}^{n}\frac{\|\nabla_{\mathbf{w}}f_i\|_F}{\|\mathbf{x}_i\|_2}.
$$

(38)

Now the Cauchy-Schwarz inequality tells us that

$$
\left(\sum_{i=1}^{n}\frac{\|\nabla_{\mathbf{w}}f_i\|}{\|\mathbf{x}_i\|_2}\right)^2 \leq \left(\sum_{i=1}^{n}\frac{1}{\|\mathbf{x}_i\|_2^2}\right)\cdot\left(\sum_{i=1}^{n}\|\nabla_{\mathbf{w}}f_i\|_F^2\right).
$$

(39)

Therefore

$$
\mathrm{MLS} \leq \|\mathbf{W}\|_2\sqrt{\frac{1}{n}\sum_{i=1}^{n}\frac{1}{\|\mathbf{x}_i\|_2^2}} \cdot \sqrt{\frac{1}{n}\sum_{i=1}^{n}\|\nabla_{\mathbf{w}}f_i\|_F^2}
$$

$$
\leq \|\mathbf{W}\|_2\sqrt{\frac{1}{n}\sum_{i=1}^{n}\frac{1}{\|\mathbf{x}_i\|_2^2}} \cdot S(\boldsymbol{\theta}^*)^{1/2}.
$$

(40)

$\square$

Now we can prove Proposition 3.10.

**Proposition.** *The network maximum local sensitivity is upper bounded by a sharpness related quantity:*

$$
\mathrm{NMLS} = \frac{1}{n}\sum_{l=1}^{L}\sum_{i=1}^{n}\|\nabla_{\mathbf{x}^l}f^l(\mathbf{x}_i^l,\boldsymbol{\theta}^*)\|_2 \leq \sqrt{\frac{1}{n}\sum_{i=1}^{n}\sum_{l=1}^{L}\frac{\|\mathbf{W}_l\|_2^2}{\|\mathbf{x}_i^l\|^2}} \cdot S(\boldsymbol{\theta}^*)^{1/2}.
$$

(41)

*Proof.* We can apply Equation (40) to every linear layer and again apply the Cauchy-Schwarz inequality to obtain

$$
\begin{aligned}
\text{NMLS} &= \frac{1}{n} \sum_{l=1}^{L} \sum_{i=1}^{n} \|\nabla_{\mathbf{x}} f_l(\mathbf{x}_i^l, \boldsymbol{\theta}^*)\|_2 \\
&\leq \sum_{l=1}^{L} \left( \sqrt{\frac{1}{n} \sum_{i=1}^{n} \frac{\|\mathbf{W}_l\|_2^2}{\|\mathbf{x}_i^l\|_2^2}} \sqrt{\frac{1}{n} \sum_{i=1}^{n} \|\nabla_{\mathbf{w}_l} f_i^l\|_F^2} \right) \\
&\leq \sqrt{\frac{1}{n} \sum_{i=1}^{n} \sum_{l=1}^{L} \frac{\|\mathbf{W}_l\|_2^2}{\|\mathbf{x}_i^l\|_2^2}} \sqrt{\frac{1}{n} \sum_{i=1}^{n} \sum_{l=1}^{L} \|\nabla_{\mathbf{w}_l} f_i^l\|_F^2} \\
&\leq \sqrt{\frac{1}{n} \sum_{i=1}^{n} \sum_{l=1}^{L} \frac{\|\mathbf{W}_l\|_2^2}{\|\mathbf{x}_i^l\|_2^2}} \cdot S(\boldsymbol{\theta}^*)^{1/2}.
\end{aligned}
\tag{42}
$$

Note that the gap in the last inequality is significantly smaller than that of Equation (40) since now we consider all linear weights. □

## F    Reparametrization-invariant sharpness and input-invariant MLS

In this section, we demonstrate that various reparametrization-invariant sharpness metrics proposed in the literature can be understood as measuring robustness of outputs to inputs and internal representations through MLS-like quantities. As discussed in Section 3.4, this provides a novel perspective: reparametrization-invariant sharpness is fundamentally characterized by the robustness of outputs to internal neural representations. We examine two different approaches from the literature—Tsuzuku et al. (2019)'s matrix-normalized and normalized sharpness (Section F.1), and Kwon et al. (2021); Andriushchenko et al. (2023)'s elementwise-adaptive sharpness (Section F.2)—showing how each relates to different variants of MLS.

### F.1    Reparametrization-invariant sharpness in Tsuzuku et al. (2019)

In this appendix, we show that the reparametrization-invariant sharpness metrics introduced in Tsuzuku et al. (2019) can be seen as an effort to tighten the bound that we derived above. We formalize two metrics from their work and establish their connections to MLS.

**Definition F.1** (Matrix-normalized sharpness (Tsuzuku et al., 2019)). *The **matrix-normalized sharpness** of a neural network is defined as*

$$
S_{matrix}(\boldsymbol{\theta}) = \sum_{l=1}^{L} \|\mathbf{W}_l\|_2 \sqrt{S(\mathbf{W}_l)},
\tag{43}
$$

*where $S(\mathbf{W}_l)$ is the trace of Hessian of the loss w.r.t. the weights of the l-th layer, and $\|\mathbf{W}_l\|_2$ is the spectral norm of the weight matrix at layer l.*

**Proposition F.2.** *The matrix-normalized sharpness upper-bounds weighted MLS across layers:*

$$
\sum_{l=1}^{L} \overline{\mathbf{x}}^l \cdot \text{MLS}^l \leq S_{matrix}(\boldsymbol{\theta}),
\tag{44}
$$

*where $\overline{\mathbf{x}}^l = \left( \frac{1}{n} \sum_{i=1}^{n} \frac{1}{\|\mathbf{x}_i^l\|_2^2} \right)^{-\frac{1}{2}}$ is the inverse of the average input norm at layer l, and $\text{MLS}^l$ is the MLS at layer l.*

*Proof.* From Equation (40) we have

$$
\sum_{l=1}^{L} \overline{\mathbf{x}}^l \cdot \text{MLS}^l \leq \sum_{l=1}^{L} \|\mathbf{W}_l\|_2 \sqrt{\frac{1}{n} \sum_{i=1}^{n} \|\nabla_{\mathbf{w}_l} f_i^l\|_F^2} \approx \sum_{l=1}^{L} \|\mathbf{W}_l\|_2 \sqrt{S(\mathbf{W}_l)} = S_{\text{matrix}}(\boldsymbol{\theta}).
\tag{45}
$$

Note that a similar inequality holds if we use Frobenius norm instead of 2-norm of the weights. □

**Definition F.3** (Normalized sharpness (Tsuzuku et al., 2019))**.** *The **normalized sharpness** of a neural network is defined as the solution to the optimization problem:*

$$S_{normalized}(\boldsymbol{\theta}) = \min_{\boldsymbol{\sigma},\boldsymbol{\sigma}'} \sum_{i,j} \left( \frac{\partial^2 L}{\partial W_{i,j} \partial W_{i,j}} (\sigma_i \sigma_j')^2 + \frac{W_{i,j}^2}{4\lambda^2 (\sigma_i \sigma_j')^2} \right), \tag{46}$$

*where $\boldsymbol{\sigma}, \boldsymbol{\sigma}'$ are scaling vectors and $\lambda$ is a hyperparameter.*

**Proposition F.4.** *The normalized sharpness optimization problem is equivalent to minimizing an upper bound on a scale-invariant MLS-like quantity:*

$$S_{normalized}(\boldsymbol{\theta}) \geq \frac{1}{\lambda} \|\mathrm{diag}(\boldsymbol{\sigma}'^{-1}) \nabla_{\mathbf{x}} f\|_F \|\mathrm{diag}(\boldsymbol{\sigma}')\mathbf{x}\|_2, \tag{47}$$

*where the quantity on the right is invariant under the transformation $\mathbf{W}\mathbf{x} \to \mathbf{W}\mathrm{diag}(\boldsymbol{\sigma}^{-1})(\mathrm{diag}(\boldsymbol{\sigma})\mathbf{x})$.*

*Proof.* Note that by Lemma C.1, $\frac{\partial^2 L}{\partial W_{i,j} \partial W_{i,j}} \approx \|\nabla_{\mathbf{W}_{i,j}} f\|_2^2$. Moreover, we have

$$\sum_{i,j} \left( \|\nabla_{\mathbf{W}_{i,j}} f\|^2 (\sigma_i \sigma_j')^2 + \frac{W_{i,j}^2}{4\lambda^2 (\sigma_i \sigma_j')^2} \right) \geq \frac{1}{\lambda} \sqrt{\sum_{i,j} (\nabla_{\mathbf{W}_{i,j}} f)^2 (\sigma_i \sigma_j')^2} \cdot \sqrt{\sum_{i,j} \frac{W_{i,j}^2}{(\sigma_i \sigma_j')^2}}$$

$$\geq \frac{1}{\lambda} \|\mathrm{diag}(\boldsymbol{\sigma}) J\|_F \|\mathrm{diag}(\boldsymbol{\sigma}')\mathbf{x}\|_2 \|\mathrm{diag}(\boldsymbol{\sigma}^{-1})\mathbf{W}\mathrm{diag}(\boldsymbol{\sigma}'^{-1})\|_F \tag{48}$$

$$\geq \frac{1}{\lambda} \|\mathrm{diag}(\boldsymbol{\sigma}'^{-1}) W^T J\|_F \|\mathrm{diag}(\boldsymbol{\sigma}')\mathbf{x}\|_2$$

$$= \frac{1}{\lambda} \|\mathrm{diag}(\boldsymbol{\sigma}'^{-1}) \nabla_{\mathbf{x}} f\|_F \|\mathrm{diag}(\boldsymbol{\sigma}')\mathbf{x}\|_2,$$

where $J = \frac{\partial f(\mathbf{W}\mathbf{x};\bar{\boldsymbol{\theta}})}{\partial (\mathbf{W}\mathbf{x})}$ (see some of the calculations in Equation (4)). Therefore, the optimization problem Equation (46) is equivalent to choosing $\boldsymbol{\sigma}, \boldsymbol{\sigma}'$ to minimize the upper bound on a scale-invariant MLS-like quantity (the quantity is invariant under the transformation of the first layer from $\mathbf{W}\mathbf{x}$ to $\mathbf{W}\mathrm{diag}(\boldsymbol{\sigma}^{-1})(\mathrm{diag}(\boldsymbol{\sigma})\mathbf{x})$, where $\mathrm{diag}(\boldsymbol{\sigma})\mathbf{x}$ becomes the new input). □

For simplicity, we do not scale the original dataset in our work and only compare MLS within the same dataset. As a result, we can characterize those reparametrization-invariant sharpness metrics by the robustness of output to the input. If we consider all linear weights in the network, then those metrics indicate the robustness of output to internal network representations.

In summary, we have shown that Tsuzuku et al. (2019)'s reparametrization-invariant sharpness metrics can be interpreted through our MLS framework: their matrix-normalized sharpness corresponds exactly to the upper bound on weighted MLS across layers, while their normalized sharpness minimizes a scale-invariant MLS-like quantity. This establishes that these metrics, while designed for reparametrization invariance, are fundamentally measuring the robustness of outputs to inputs and internal representations.

### F.2 Reparametrization-invariant sharpness upper-bounds input-invariant MLS

Having shown that Tsuzuku et al. (2019)'s metrics tighten our standard MLS/NMLS bounds, we now examine a different approach to reparametrization-invariant sharpness. In this subsection, we prove a formal result relating elementwise-adaptive sharpness from Kwon et al. (2021); Andriushchenko et al. (2023) to a different variant of MLS: input-invariant MLS. This demonstrates that different reparametrization-invariant sharpness metrics from the literature can be understood as measuring robustness of outputs to inputs through different MLS variants.

In this appendix, we consider the adaptive average-case n-sharpness considered in Kwon et al. (2021); Andriushchenko et al. (2023):

$$S_{\text{avg}}^{\rho}(\mathbf{w}, |\mathbf{w}|) \triangleq \frac{2}{\rho^2} \mathbb{E}_{S \sim P_n, \delta \sim \mathcal{N}(0, \rho^2 \text{diag}(|\mathbf{w}|^2))} \left[ L_S(\mathbf{w} + \delta) - L_S(\mathbf{w}) \right], \tag{49}$$

which is shown to be *elementwise* adaptive sharpness in Andriushchenko et al. (2023). They also show that for a thrice differentiable loss, $L(w)$, the average-case elementwise adaptive sharpness can be written as

$$S_{\text{avg}}^{\rho}(\mathbf{w}, |\mathbf{w}|) = \mathbb{E}_{S \sim P_n} \left[ \text{Tr} \left( \nabla^2 L_S(\mathbf{w}) \odot |\mathbf{w}| |\mathbf{w}|^{\top} \right) \right] + O(\rho). \tag{50}$$

**Definition F.5.** *We define the **Elementwise-Adaptive Sharpness** $S_{adaptive}$ to be*

$$S_{adaptive}(\mathbf{w}) \triangleq \lim_{\rho \to 0} S_{avg}^{\rho}(\mathbf{w}, |\mathbf{w}|) = \mathbb{E}_{S \sim P_n} \left[ \text{Tr} \left( \nabla^2 L_S(\mathbf{w}) \odot |\mathbf{w}| |\mathbf{w}|^{\top} \right) \right] \tag{51}$$

In this appendix, we focus on the property of $S_{\text{adaptive}}$ instead of the approximation Equation (50). Adapting the proof of *Lemma C.1*, we have the following lemma.

**Lemma F.6.** *If $\boldsymbol{\theta}$ is an approximate interpolation solution, i.e. $\|f(\mathbf{x}_i, \boldsymbol{\theta}) - \mathbf{y}_i\| < \varepsilon$ for $i \in \{1, 2, \cdots, n\}$, $|\boldsymbol{\theta}_j|^2 \|\nabla_{\theta_j}^2 f(\mathbf{x}_i, \boldsymbol{\theta})\| < M$ for all $j$, and $L$ is MSE loss, then*

$$S_{adaptive}(\boldsymbol{\theta}^*) = \frac{1}{n} \sum_{i=1}^{n} \sum_{j=1}^{m} |\boldsymbol{\theta}_j|^2 \left\| \nabla_{\boldsymbol{\theta}_j} f(\mathbf{x}_i, \boldsymbol{\theta}) \right\|_2^2 + O(\varepsilon), \tag{52}$$

*where $m$ is the number of parameters.*

*Proof.* Using basic calculus we get

$$\begin{aligned}
S_{\text{adaptive}}(\boldsymbol{\theta}) &= \frac{1}{2n} \sum_{i=1}^{n} \text{Tr}(\nabla_{\boldsymbol{\theta}}^2 \|f(\mathbf{x}_i, \boldsymbol{\theta}) - \mathbf{y}_i\|^2 \odot |\boldsymbol{\theta}| |\boldsymbol{\theta}|^{\top}) \\
&= \frac{1}{2n} \sum_{i=1}^{n} \text{Tr} \nabla_{\boldsymbol{\theta}} (2(f(\mathbf{x}_i, \boldsymbol{\theta}) - \mathbf{y}_i)^T \nabla_{\boldsymbol{\theta}} f(\mathbf{x}_i, \boldsymbol{\theta})) \odot |\boldsymbol{\theta}| |\boldsymbol{\theta}|^{\top} \\
&= \frac{1}{n} \sum_{i=1}^{n} \sum_{j=1}^{m} |\boldsymbol{\theta}_j|^2 \frac{\partial}{\partial \boldsymbol{\theta}_j} ((f(\mathbf{x}_i, \boldsymbol{\theta}) - \mathbf{y}_i)^T \nabla_{\boldsymbol{\theta}} f(\mathbf{x}_i, \boldsymbol{\theta}))_j \\
&= \frac{1}{n} \sum_{i=1}^{n} \sum_{j=1}^{m} |\boldsymbol{\theta}_j|^2 \frac{\partial}{\partial \boldsymbol{\theta}_j} (f(\mathbf{x}_i, \boldsymbol{\theta}) - \mathbf{y}_i)^T \nabla_{\boldsymbol{\theta}_j} f(\mathbf{x}_i, \boldsymbol{\theta}) \\
&= \frac{1}{n} \sum_{i=1}^{n} \sum_{j=1}^{m} |\boldsymbol{\theta}_j|^2 \left\| \nabla_{\boldsymbol{\theta}_j} f(\mathbf{x}_i, \boldsymbol{\theta}) \right\|_2^2 + |\boldsymbol{\theta}_j|^2 (f(\mathbf{x}_i, \boldsymbol{\theta}) - \mathbf{y}_i)^T \nabla_{\theta_j}^2 f(\mathbf{x}_i, \boldsymbol{\theta}) \\
&= \frac{1}{n} \sum_{i=1}^{n} \sum_{j=1}^{m} |\boldsymbol{\theta}_j|^2 \left\| \nabla_{\boldsymbol{\theta}_j} f(\mathbf{x}_i, \boldsymbol{\theta}) \right\|_2^2 + \frac{1}{n} \sum_{i=1}^{n} \sum_{j=1}^{m} |\boldsymbol{\theta}_j|^2 (f(\mathbf{x}_i, \boldsymbol{\theta}) - \mathbf{y}_i)^T \nabla_{\theta_j}^2 f(\mathbf{x}_i, \boldsymbol{\theta})
\end{aligned}$$

Therefore

$$\left| S_{\text{adaptive}}(\boldsymbol{\theta}) - \frac{1}{n} \sum_{i=1}^{n} \sum_{j=1}^{m} |\boldsymbol{\theta}_j|^2 \left\| \nabla_{\boldsymbol{\theta}_j} f(\mathbf{x}_i, \boldsymbol{\theta}) \right\|_2^2 \right| < \frac{1}{n} \sum_{i=1}^{n} \sum_{j=1}^{m} |(f(\mathbf{x}_i, \boldsymbol{\theta}) - \mathbf{y}_i)^T |\boldsymbol{\theta}_j|^2 \nabla_{\theta_j}^2 f(\mathbf{x}_i, \boldsymbol{\theta})| < mM\varepsilon = O(\varepsilon). \tag{53}$$

$\square$

**Definition F.7.** *We define the **Input-invariant MLS** of a network $f : \mathbb{R}^M \to \mathbb{R}^N$ to be*

$$\frac{1}{n} \sum_{i=1}^{n} \sum_{p=1}^{N} \left\| \nabla_{x_p^{(i)}} f \right\|_2^2 (x_p^{(i)})^2 \; , \tag{54}$$

*where $x_p^{(i)}$ is the p-th entry of i-th training sample.*

It turns out that again the adaptive sharpness upper bounds the input-invariant MLS.

**Proposition F.8.** *Assuming that the condition of Lemma F.6 holds, then elementwise-adaptive sharpness upper-bounds input-invariant MLS:*

$$\frac{1}{n} \sum_{i=1}^{n} \sum_{j=1}^{m} |\boldsymbol{\theta}_j|^2 \left\| \nabla_{\boldsymbol{\theta}_j} f(\mathbf{x}_i, \boldsymbol{\theta}) \right\|_2^2 \geq \frac{1}{nd} \sum_{i=1}^{n} \sum_{p=1}^{N} \left\| \nabla_{x_p^{(i)}} f \right\|_2^2 (x_p^{(i)})^2 \tag{55}$$

*Proof.* Now we adapt the linear stability trick. For $\boldsymbol{\theta} = \mathbf{W}$, the first-layer weights, we have

$$
\begin{aligned}
\sum_{j=1}^{m} |\boldsymbol{\theta}_j|^2 \left\| \nabla_{\boldsymbol{\theta}_j} f(\mathbf{x}, \boldsymbol{\theta}) \right\|_2^2 &= \sum_{i,j,k} J_{jk}^2 \mathbf{W}_{ki}^2 x_p^2 \\
&= \sum_{i,j} \left( \sum_{k=1}^{d} J_{jk}^2 \mathbf{W}_{ki}^2 \right) x_p^2 \\
&\geq \frac{1}{d} \sum_i \left\| \nabla_{x_p} f \right\|_2^2 x_p^2
\end{aligned}
\tag{56}
$$

where same as in Equation (4), $J = \frac{\partial f(\mathbf{W}\mathbf{x}; \bar{\boldsymbol{\theta}})}{\partial (\mathbf{W}\mathbf{x})}$, $\nabla_{\mathbf{x}} f(\mathbf{W}\mathbf{x}; \bar{\boldsymbol{\theta}}) = J\mathbf{W}$, and $x_p$ is the p-th entry of $\mathbf{x}$. Taking the sample mean of both sides proves the proposition.

$\square$

# G   Numerical approximation of normalized MLS and elementwise-adaptive sharpness

In this appendix, we detail how we approximate the normalized MLS and adaptive sharpness in Section 4.3. Note that for all network $f$ the last layer is the sigmoid function, so the output is bounded in $(0, 1)$, and we use MSE loss to be consistent with the rest of the paper.

For the adaptive sharpness, we adopt the definition in Andriushchenko et al. (2023) and use the sample mean to approximate the expectation in Equation (49). Therefore, for network $f(\mathbf{w})$,

$$S_{\text{adaptive}}(f) = \frac{1}{nm} \sum_{i=1}^{n} \sum_{j=1}^{m} L(\mathbf{x}_i; \mathbf{w} + \delta_j) - L(\mathbf{x}_i; \mathbf{w}), \tag{57}$$

where $\delta \sim \mathcal{N}(0, 0.01 \, \text{diag}(|w|^2))$.

For normalized MLS, we first reiterate the definition from the main text. We use normalized MLS below as an approximation to the input-invariant MLS (Definition F.7), because the latter is computationally prohibitive for modern large ViTs. On the other hand, there is an efficient way to estimate normalized MLS as detailed below.

**Definition G.1.** *We define the **normalized MLS** as $\frac{1}{n} \sum_{i=1}^{n} \|\mathbf{x}_i\|_2^2 \|\nabla_{\mathbf{x}_i} f\|_2^2$*

Therefore, to approximate normalized MLS, we need to approximate $\|\nabla_{\mathbf{x}_i} f\|_2$. By definition of matrix 2-norm,

$$\|\nabla_{\mathbf{x}} f\|_2 = \sup_{\delta} \frac{\|\nabla_{\mathbf{x}} f \, \delta\|_2}{\|\delta\|_2} \approx \max_{\delta} \frac{\|f(\mathbf{x} + \delta) - f(\mathbf{x})\|_2}{\|\delta\|_2}. \tag{58}$$

To solve this optimization problem, we start from a randomly sampled vector $\delta$ that has the same shape as the network input, and we update $\delta$ using gradient descent.

# H   Empirical analysis of the bound

## H.1   Tightness of the bound

In this section, we mainly explore the tightness of the bound in Equation (11) for reasons discussed in Section 3.2. First we rewrite Equation (11) as

$$
\begin{aligned}
\mathrm{MLS} = \frac{1}{n}\sum_{i=1}^{n}\|\nabla_{\mathbf{x}}f(\mathbf{x}_i,\boldsymbol{\theta}^*)\|_2 \qquad &:= A \\
\leq \frac{\|\mathbf{W}\|_2}{n}\sum_{i=1}^{n}\frac{\|\nabla_{\mathbf{W}}f(\mathbf{x}_i,\boldsymbol{\theta}^*)\|_F}{\|\mathbf{x}_i\|_2} \qquad &:= B \\
\leq \|\mathbf{W}\|_2\sqrt{\frac{1}{n}\sum_{i=1}^{n}\frac{1}{\|\mathbf{x}_i\|_2^2}}\sqrt{\frac{1}{n}\sum_{i=1}^{n}\|\nabla_{\mathbf{W}}f(\mathbf{x}_i,\boldsymbol{\theta}^*)\|_F^2} \qquad &:= C \\
\leq \|\mathbf{W}\|_2\sqrt{\frac{1}{n}\sum_{i=1}^{n}\frac{1}{\|\mathbf{x}_i\|_2^2}}S(\boldsymbol{\theta}^*)^{1/2} \qquad &:= D
\end{aligned}
\tag{59}
$$

Thus Equation (11) consists of 3 different steps of relaxations. We analyze them one by one:

1. $(A \leq B)$ The equality holds when $\|W^T J\|_2 = \|W\|_2\|J\|_2$ and $\|J\|_F = \|J\|_2$, where $J = \frac{\partial f(\mathbf{W}\mathbf{x};\bar{\boldsymbol{\theta}})}{\partial(\mathbf{W}\mathbf{x})}$. The former equality requires that $W$ and $J$ have the same left singular vectors. The latter requires $J$ to have zero singular values except for the largest singular value. Since $J$ depends on the specific neural network architecture and training process, we test the tightness of this bound empirically (Figure H.6).

2. $(B \leq C)$ The equality requires $\frac{\|\nabla_{\mathbf{W}}f(\mathbf{x}_i,\boldsymbol{\theta}^*)\|_F}{\|\mathbf{x}_i\|_2}$ to be the same for all $i$. In other words, the bound is tight when $\frac{\|\nabla_{\mathbf{W}}f(\mathbf{x}_i,\boldsymbol{\theta}^*)\|_F}{\|\mathbf{x}_i\|_2}$ does not vary too much from sample to sample.

3. $(C \leq D)$ The equality holds if the model is linear, i.e. $\boldsymbol{\theta} = \mathbf{W}$.

We empirically verify the tightness of the above bounds in Figure H.6

## H.2   Correlation analysis

We empirically show how different metrics correlate with each other, and how these correlations can be predicted from our bounds. We train 100 VGG-11 networks with different batch sizes, learning rates, and random initialization to classify images from the CIFAR-10 dataset, and plot pairwise scatter plots between different quantities at the end of the training: local dimensionality, sharpness (square root of Equation (3)), log volume (Equation (7)), MLS (Equation (11)), NMLS (Equation (12)), generalization gap (gen gap), D (Equation (59)), bound (right-hand side of Equation (12)) and relative sharpness (Petzka et al., 2021) (see Figure H.7). We only include CIFAR-10 data with 2 labels to ensure that the final training accuracy is close to 100%.

We repeat the analysis on MLPs and LeNets trained on the FashionMNIST dataset and the CIFAR-10 dataset (Figure H.8 and Figure H.9). We find that

1. The bound over NMLS, MLS, and NMLS introduced in Equation (12) and Equation (11) consistently correlates positively with the generalization gap.

2. Although the bound in Equation (9) is loose, log volume correlates well with sharpness and MLS.

3. Sharpness is positively correlated with the generalization gap, indicating that little reparametrization effect (Dinh et al., 2017) is happening during training, i.e. the network weights do not change too much during training. This is consistent with observations in Ma & Ying (2021).

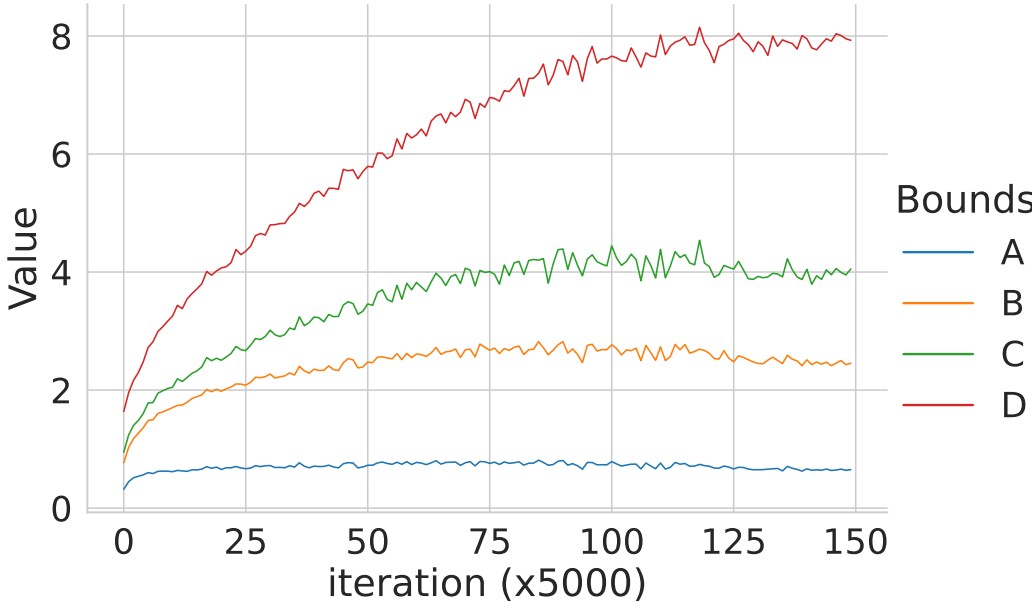

Figure H.6: **Empirical tightness of the bounds.** We empirically verify that the inequalities in Equation (59) hold and test their tightness. The results are shown for a fully connected feedforward network trained on the FashionMNIST dataset. The quantities A, B, C, and D are defined in Equation (59). We see that the gap between C and D is large compared to the gap between A and B or B and C. This indicates that partial sharpness $\|\nabla_{\mathbf{W}} f(\mathbf{x}_i, \boldsymbol{\theta}^*)\|_F$ (sensitivity of the loss w.r.t. only the input weights) is more indicative of the change in the maximum local sensitivity (A). Indeed, correlation analysis shows that bound C is positively correlated with MLS while bound D, perhaps surprisingly, is negatively correlated with MLS (Figure H.8).

4. The bound derived in Equation (12) correlates positively with NMLS in all experiments.

5. MLS that only consider the first layer weights can sometimes negatively correlate with the bound derived in Equation (11) (Figure H.8).

6. Relative flatness that only consider the last layer weights introduced in (Petzka et al., 2021) shows weak (even negative) correlation with the generalization gap. Note that "relative flatness" is a misnomer that is easier understood as "relative *sharpness*", and is supposed to be *positively* correlated with the generalization gap.

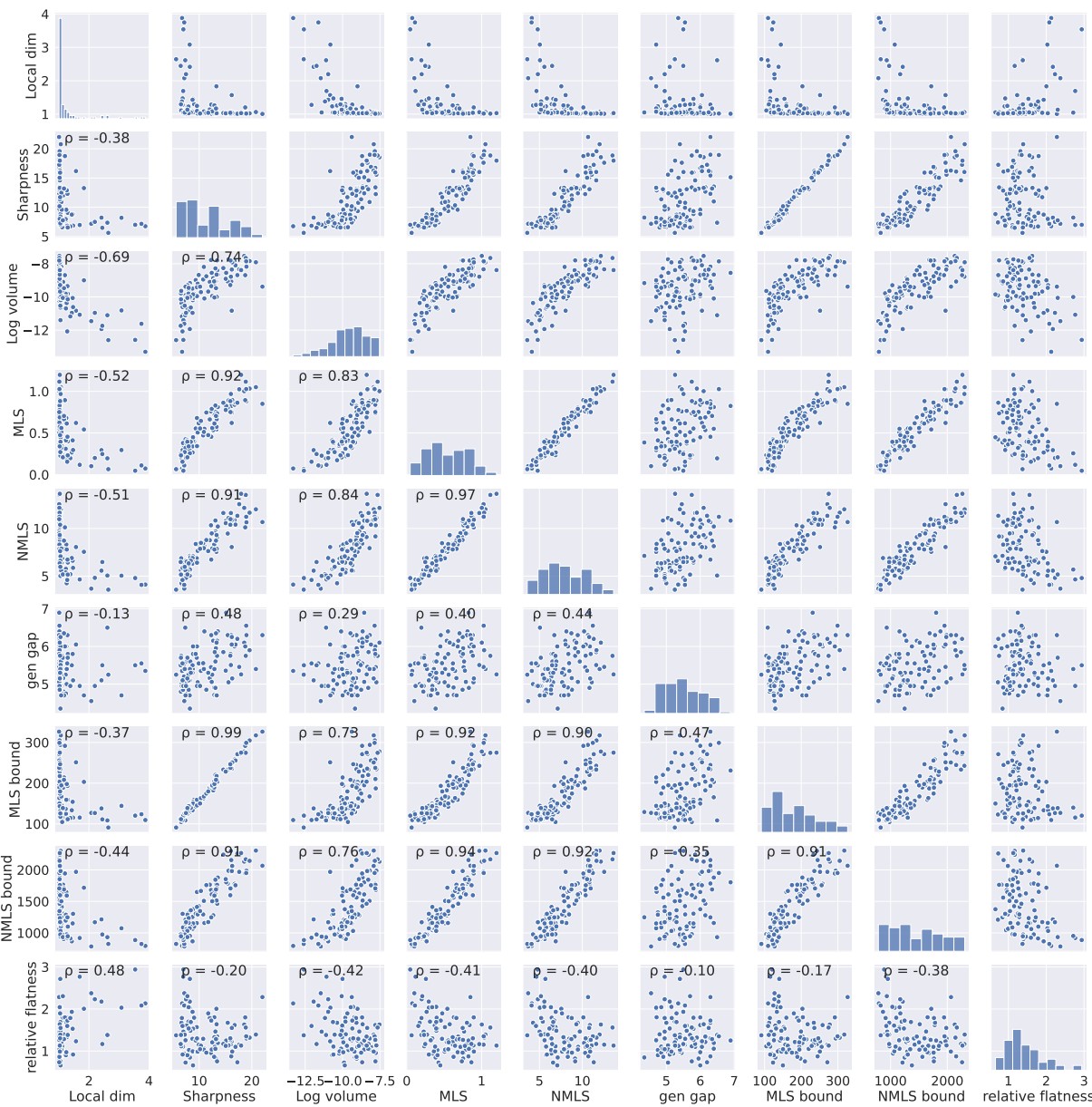

Figure H.7: **Pairwise correlation among different metrics.** We trained 100 different VGG-11 networks on the CIFAR-10 dataset using vanilla SGD with different learning rates, batch sizes, and random initializations and plot pairwise scatter plots between different quantities: local dimensionality, sharpness (square root of Equation (3)), log volume (Equation (7)), MLS (Equation (11)), NMLS (Equation (12)), generalization gap (gen gap), MLS bound (Proposition 3.8), NMLS bound (Proposition 3.10) and relative sharpness ((Petzka et al., 2021)). The Pearson correlation coefficient $\rho$ is shown in the top-left corner for each pair of quantities. See Appendix H.2 for a summary of the findings in this figure.

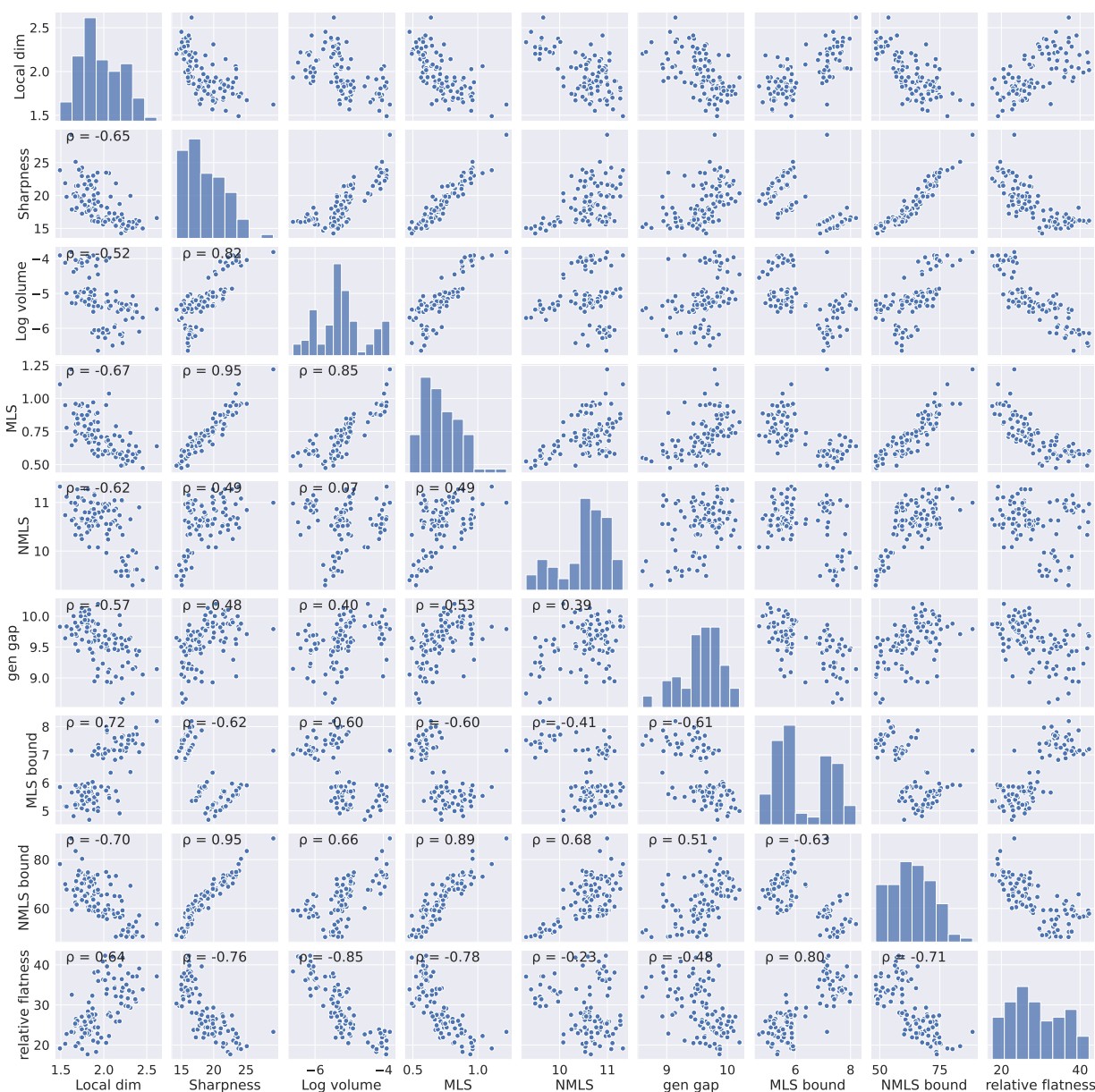

Figure H.8: **Pairwise correlation among different metrics.** We trained 100 different 4-layer MLPs on the FashionMNIST dataset using vanilla SGD with different learning rates, batch size, and random initializations and plot pairwise scatter plots between different quantities: local dimensionality, sharpness (square root of Equation (3)), log volume (Equation (7)), MLS (Equation (11)), NMLS (Equation (12)), generalization gap (gen gap), MLS bound (Proposition 3.8), NMLS bound (Proposition 3.10) and relative sharpness ((Petzka et al., 2021)). The Pearson correlation coefficient $\rho$ is shown in the top-left corner for each pair of quantities. See Appendix H.2 for a summary of the findings in this figure.

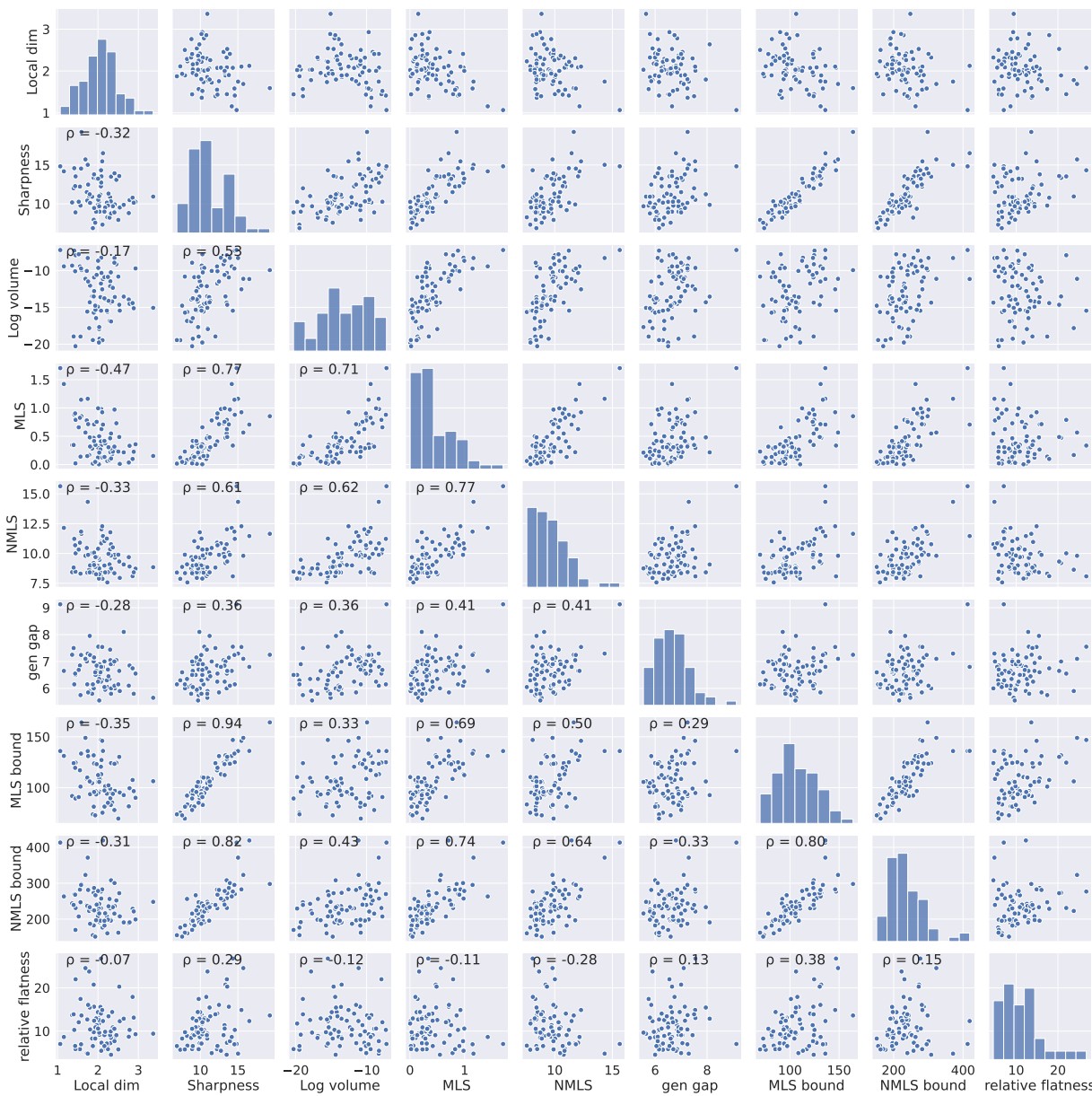

Figure H.9: **Pairwise correlation among different metrics.** We trained 100 different LeNets on the CIFAR-10 dataset using vanilla SGD with different learning rates, batch size, and random initializations and plot pairwise scatter plots between different quantities: local dimensionality, sharpness (square root of Equation (3)), log volume (Equation (7)), MLS (Equation (11)), NMLS (Equation (12)), generalization gap (gen gap), MLS bound (Proposition 3.8), NMLS bound (Proposition 3.10) and relative sharpness ((Petzka et al., 2021)). The Pearson correlation coefficient $\rho$ is shown in the top-left corner for each pair of quantities. See Appendix H.2 for a summary of the findings in this figure.

# I Additional experiments

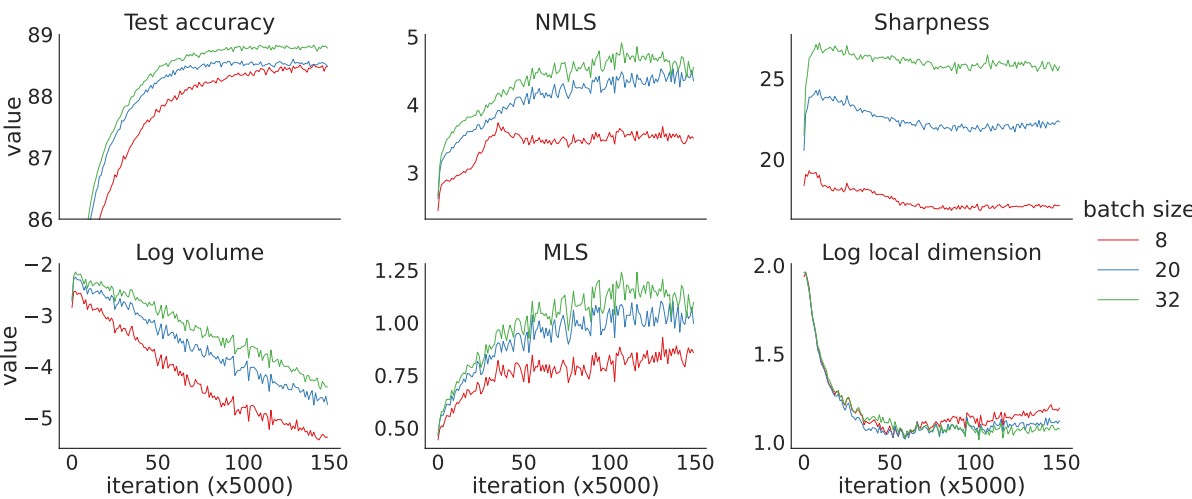

Figure I.10: Trends in key variables across SGD training of a 4-layer MLP with fixed learning rate (equal to 0.1) and varying batch size (8, 20, and 32). MLS/NMLS closely follows the trend of sharpness during the training. From left to right: test accuracy, NMLS, sharpness (square root of Equation (3)), log volumetric ratio (Equation (7)), MLS (Equation (11)), and local dimensionality of the network output (Equation (15)).

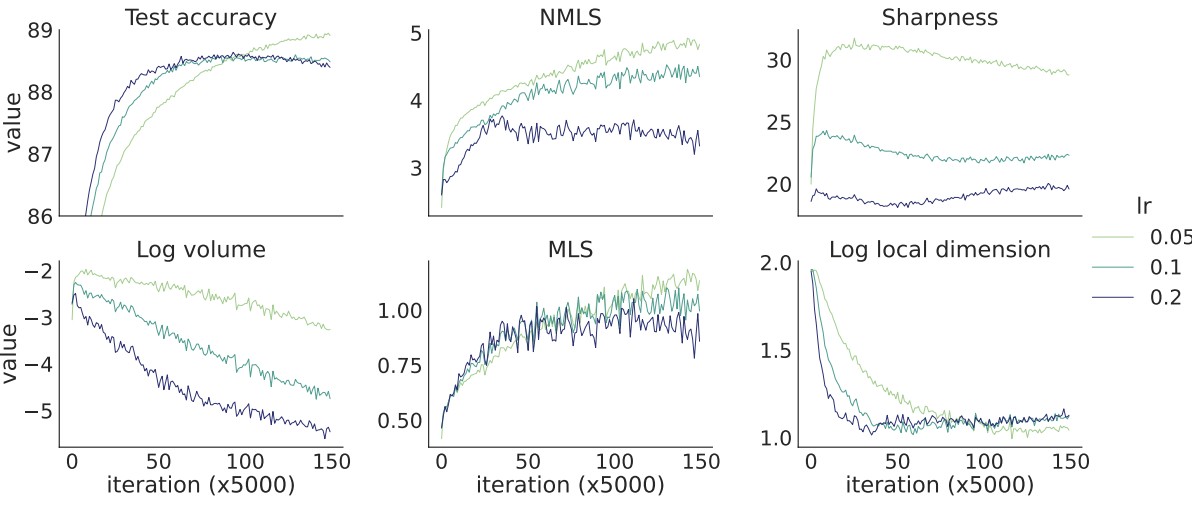

Figure I.11: Trends in key variables across SGD training of a 4-layer MLP with fixed batch size (equal to 20) and varying learning rates (0.05, 0.1 and 0.2). MLS/NMLS closely follows the trend of sharpness during the training. From left to right: test accuracy, NMLS, sharpness (square root of Equation (3)), log volumetric ratio (Equation (7)), MLS (Equation (11)), and local dimensionality of the network output (Equation (15)).

## J    Sharpness and compression on test set data

Even though Equation (3) is exact for interpolation solutions only (i.e., those with zero loss), we found that the test loss is small enough (Figure J.12) so that it should be a good approximation for test data as well. Therefore we analyzed our simulations to study trends in sharpness and volume for these held-out test data as well (Figure J.12). We observed that this sharpness increased rather than diminished as a result of training. We hypothesized that sharpness could correlate with the difficulty of classifying testing points. This was supported by the fact that the sharpness of misclassified test data was even greater than that of all test data. Again we see that MLS has the same trend as the sharpness. Despite this increase in sharpness, the volume followed the same pattern as the training set. This suggests that compression in representation space is a robust phenomenon that can be driven by additional phenomena beyond sharpness. Nevertheless, the compression still is weaker for misclassified test samples that have higher sharpness than other test samples. Overall, these results emphasize an interesting distinction between how sharpness evolves for training vs. test data.

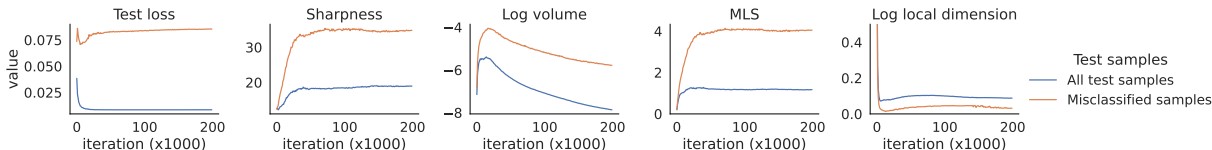

Figure J.12: Trends in key variables across SGD training of the VGG-11 network with fixed learning rate (equal to 0.1) and batch size (equal to 20) for samples of the test set. After the loss is minimized, we compute sharpness and volume on the test set. Moreover, the same quantities are computed separately over the entire test set or only on samples that are misclassified. In order from left to right in row-wise order: test loss, sharpness (Equation (2)), log volumetric ratio (Equation (7)), MLS, and local dimensionality of the network output (Equation (15)).

## K    Computational resources and code availability

All experiments can be run on one NVIDIA Quadro RTX 6000 GPU. The code is available at `https://github.com/chinsengi/sharpness-compression`.

