# OpenReview forum: "A simple connection from loss flatness to compressed neural representations"
_TMLR — Accepted by TMLR_

### Review · Reviewer_sf7P · 2025-09-10

**Summary Of Contributions:**

The role of sharpness in trained deep neural networks remains unclear in the existing literature. The authors of this work offer a different perspective on sharpness, arguing sharpness measures the degree of compression in the network's representation space. Specifically, this work argues if a deep network's parameters lie in a flatter minima, the representations are compressed to a higher degree. The authors study two compression metrics - local volumetric ratio (LVR) and (network) maximum local sensitivity ( (N)MLS ) - and derive upper bounds on the metrics that depend on sharpness. They include empirical results on a variety of architectures on image classification datasets.



**Strengths:**
1. Connecting the geometry of the loss landscape in the parameter space to the geometry of the features in the representation space is interesting, and can potentially provide deeper insights into the interpretability of trained networks.

2. The empirical results on image classification datasets are convincing in showing a clear correlation between LVR / (N)MLS and sharpness.



**Weaknesses / questions:**
1. The derived bounds are architecture-independent, and are quite loose as a result, as seen in Figures 3 and H.5.

2. The claim that the derived bounds are loss-agnostic does not seem entirely accurate (third paragraph in Section 2). The derivation of Eq. 3 depends on the use of the quadratic loss, and the proofs of Propositions 3.4, 3.6, 3.8, and 3.10 all use Eq. 3. Therefore, it seems to me the theoretical results depend on the quadratic loss.

3. Propositions 3.4, 3.6, 3.8, and 3.10 respectively provide upper bound LVR, MLS, and NMLS that depend on sharpness. However, the bounds also depend on the norms of the weight matrices themselves, which clearly depends on the solution $\boldsymbol{\theta}^\star$. Due to this dependence, it's not clear to me that a lower sharpness *always* implies more compression, as it's not obvious a decrease in sharpness from one interpolating solution to another should always "counteract" any potential increases in $||\mathbf{W}||_2$ and/or $||\mathbf{W}_l||_2$ due to the change in $\boldsymbol{\theta}^\star$. Although this seems to be the case from the empirical results, could the authors comment on this point?

4. In Section 3.5 after Eq. 16, the authors state “On the other hand, our theory then broadly applies to cases where the number of classes is much larger than the feature dimension, …, where generalized neural collapse can occur (Jiang et al., 2023).” I understand due to the dependence on $\sigma_{min}(\mathbf{W}_L)$, Eq. 16 provides a non-vacuous bound only when the number of classes is large, which is the setting where generalized neural collapse arises. However, I don’t quite understand how Eq. 16 relates to the actual phenomena of generalized neural collapse (i.e., the geometry of the last-layer classifier and the penultimate-layer features), so it is unclear to me why the authors discuss generalized neural collapse here. I would be grateful for clarification on this point of discussion.

5. A common feature compression metric used in the neural collapse literature is the trace of the per-class and global covariances of the hidden-layer representations (see Eqs. 5-7 in [1], and also [2, 3, 4]), sometimes called *within-class variability*. I’d be curious to see the relationship between within-class variability and either LVR / (N)MLS, or sharpness. For example, is there any correlation between the within-class variabilities, and the corresponding “per-class LVR / (N)MLS," i.e., LVR / (N)MLS but only calculated for samples in each class (if this is a meaningful metric to begin with)?

6. The authors focus on the trace of the Hessian at interpolating solutions as their sharpness metric, but also mention there are other measures of sharpness. Do you empirically observe the same correlation between LVR / (N)MLS and other notions of sharpness? For instance, are the predictions / logits of trained networks with lower LVR / (N)MLS more robust to noise injected into the parameters?




[1] Peng Wang, Xiao Li, Can Yaras, Zhihui Zhu, Laura Balzano, Wei Hu, & Qing Qu. “Understanding deep representation learning via layerwise feature compression and discrimination.” arXiv preprint arXiv:2311.02960, 2023.

[2] Vignesh Kothapalli, Tom Tirer, and Joan Bruna. “A neural collapse perspective on feature evolution in graph neural networks.” arXiv preprint arXiv:2307.01951, 2023.

[3] Akshay Rangamani, Marius Lindegaard, Tomer Galanti, and Tomaso A Poggio. “Feature learning in deep classifiers through intermediate neural collapse.” In International Conference on Machine Learning, 2023.

[4] Tom Tirer, Haoxiang Huang, and Jonathan Niles-Weed. “Perturbation analysis of neural collapse.” In International Conference on Machine Learning, 2023

**Audience:**

Yes

**Audience Explanation:**

Drawing a connection between the sharpness of deep neural network parameters and the network’s resulting representation space is an interesting viewpoint, which can enhance network interpretability and inspire follow-up works to advance this perspective.

**Claims And Evidence:**

Yes

**Claims Explanation:**

One minor claim that does not appear accurate is that their theoretical results are loss-independent (third paragraph in Section 2); it seems they depend on the quadratic loss (see point 2 in weaknesses / questions above). However, all other claims appear accurate and are supported with clear evidence. The authors provide proofs in the appendix, which are fairly easy to follow and appear correct. The empirical results on different architectures and datasets show a clear correlation between LVR / (N)MLS and sharpness, which support the authors’ main claims.

**Requested Changes:**

I recommend the following changes to strengthen the work, but they are not critical in my decision for recommending acceptance.

1. Clarifying the discussion between the result in Eq. 16 and neural collapse.

2. Additional experimental results on the relationship between feature compression metrics commonly studied in neural collapse literature, and either LVR / (N)MLS or sharpness. I believe this would also strengthen the connection between the authors' work and neural collapse as discussed in Section 3.5.

3. Additional experimental results showing a correlation between (N)MLS / LVR and other sharpness metrics.

See points 4 - 6 in weaknesses / questions above for further elaboration.

---

> ### Author Response · Authors · 2025-11-26
> **Thank you**
>
> We thank Reviewer sf7P for the positive assessment and insightful questions.
>
> ### **Concern 1: Bounds are Architecture-Independent and Loose**
>
> > *"The derived bounds are architecture-independent, and are quite loose as a result..."*
>
> **Response:**
> We agree and have acknowledged this limitation explicitly in multiple places:
>
> 1. **Contributions list (Page 2):** Each contribution now includes scope and limitations
> 2. **"On bound tightness" paragraph (Page 2):** Acknowledges bounds may not be tight and emphasizes role of empirical validation
> 3. **Section 3 discussion:** "On the role of weight norms in the bounds" paragraph addresses how bound looseness varies by setting
>
> We view architecture-independence as both a strength (generality) and limitation (looseness), now made explicit.
>
> ### **Concern 2: Loss-Agnostic Claim**
>
> > *"The claim that the derived bounds are loss-agnostic does not seem entirely accurate... The derivation of Eq. 3 depends on... quadratic loss."*
>
> **Response:**
> Correct. Revised Section 2 ("Scope and loss function considerations") to clarify the scope and what we previously meant by "loss-agnostic":
>
> 1. **Theoretical scope:** Analysis uses quadratic (MSE) loss, enabling the key relationship in Eq. 3 connecting sharpness to gradients.
>
> 2. **CE loss limitations:** We discuss why cross-entropy (CE) loss is problematic for Hessian-based sharpness. Per Granziol et al., large weights + low CE loss → artificially small Hessian (deceptively "flat"); Hessian rank bounded by (neurons × classes) causes near-zero eigenvalues. This breaks the flatness-generalization connection for CE.
>
> 3. **Two pathways for non-MSE networks:**
>    - **Theoretical:** Results extend to logistic loss with label smoothing (same minimizers as MSE; Wen et al. Lemma A.13)
>    - **Practical:** Metrics (LVR, MLS, NMLS) depend on network function, not training loss. Compute sharpness using MSE post-hoc (ViT experiments, Figure 4: 181 CE-trained models)
>
> ### **Concern 3: Lower Sharpness and Compression Relationship**
>
> > *"Due to this dependence [on weight norms], it's not clear... that a lower sharpness always implies more compression... could the authors comment on this?"*
>
> **Response:**
> This is an excellent point. We have added a paragraph "On the role of weight norms in the bounds" (Section 3) that addresses this:
>
> - **Theoretical ambiguity acknowledged:** Lower sharpness might not counteract increases in prefactor terms
> - **Key distinction added:**
>   - **Single architecture during training:** Prefactors stable → sharpness dominates → strong correlation
>   - **Across different architectures:** Prefactors vary → need adaptive sharpness & normalized MLS (as in ViT experiments)
> - **References Appendix H** for detailed empirical analysis of bound tightness
> - **Conclusion:** Bounds capture meaningful relationships in practice even when not tight equalities
>
> ### **Concern 4: Neural Collapse Connection Unclear**
>
> > *"I don't quite understand how Eq. 16 relates to... generalized neural collapse (i.e., the geometry of the last-layer classifier and... features), so it is unclear... why the authors discuss generalized neural collapse here."*
>
> **Response:**
> We have revised the neural collapse discussion (Section 3.5) to be more clear and direct:
>
> 1. **Focused on what the math shows:** Bound is effective when K ≥ d (number of classes ≥ feature dimension)
> 2. **Listed concrete domains:** Language modeling, retrieval, face recognition where neural collapse occurs
> 3. **Added within-class connection:** View within-class samples as perturbations around class mean; when within-class variance vanishes (neural collapse), all samples map to same features → extreme robustness our bounds capture locally
> 4. **Added explicit caveat:** "the precise relationship between our local robustness bounds and the global geometric structure of neural collapse... remains an open question for future investigation." Connection is suggestive but not fully developed.
>
> ### **Concern 5: Correlation with Within-Class Variability**
>
> > *"I'd be curious to see the relationship between within-class variability and either LVR / (N)MLS, or sharpness."*
>
> **Response:**
> We appreciate this suggestion. However, given the two-week rebuttal timeline and the substantial effort required to conduct these studies properly, we have not included them in this revision but consider them valuable for future work. However, the revised neural collapse discussion (above) addresses the conceptual connection between within-class variability and our metrics.
>
> ### **Concern 6: Correlation with Other Sharpness Metrics**
>
> > *"Do you empirically observe the same correlation between LVR / (N)MLS and other notions of sharpness?"*
>
> **Response:**
> Focused this revision on trace-based sharpness metric; a complete analysis of its mathematical implications and empirical testing was already extensive. Exploring other sharpness metrics is of significant interest and an important direction for future investigation.

---

### Review · Reviewer_KMtE · 2025-10-12

**Summary Of Contributions:**

This paper examines the relationship between sharpness (loss of flatness around minima) and the compression of neural representations. The authors define/measure three local representation metrics—Local Volumetric Ratio (LVR), Maximum Local Sensitivity (MLS), and Network MLS (NMLS)—and prove upper bounds that tie these to sharpness (e.g., via Tr(H), Eq. (3), and Props. 3.4/3.8/3.10). Empirically, they report positive correlations between sharpness and compression metrics across MLP/CNN/ViT models, with some caveats and outliers, and discuss extensions to reparameterization-invariant sharpness.

## Strengths:
Clear technical target: Connects parameter-space geometry (sharpness) to feature-space geometry (compression), which is interesting and timely.
Concrete metrics: LVR/MLS/NMLS are well-defined and computable; NMLS is a sensible layer-aware refinement.
Theoretical bounds: Inequalities that upper-bound compression by sharpness are simple and readable.

## Weaknesses:
Clarity/flow is below bar. The abstract introduces “compression” and “reparameterization-invariant sharpness” with no setup or definitions, and the contribution bullets are stated as math without intuition or implications. The narrative arc is weak—there’s no clear thread connecting the problem, the proposed metrics, and the results. Core terms (“reparameterization,” “short description length models”) aren’t defined up front. Section 3—the main technical section—is poorly organized: definitions/lemmas/propositions arrive without motivation; Eq. (7) appears abruptly, later tied to Eq. (14), and its link to Def. 3.1 is unclear. The paper lacks an orienting figure to explain sharpness vs. flatness and how the metrics behave. In short, the exposition (almost the entire paper, especially the abstract and contribution list) needs a substantial rewrite to foreground definitions and implications.

**Audience:**

Yes

**Audience Explanation:**

The sharpness–compression connection is timely and relevant to TMLR readers (optimization, generalization, representation learning).
New metrics (LVR/MLS/NMLS) and a reparameterization-invariant angle could spark discussion, follow-up work, and tooling—even if the current exposition is below bar.

**Claims And Evidence:**

No

**Claims Explanation:**

Primarily because the paper is not presented in a proper, comprehensive way, almost the entire paper requires a rewrite, specifically the abstract and introduction. The contribution list is math-only with no intuition, scope, or implications. Section 3 interleaves definitions/lemmas/propositions without motivation or stated assumptions. I can’t verify the claims even if they were true.

**Requested Changes:**

**A bstract rewrite:** Clearly state the problem first; define “compression” and “reparameterization-invariant sharpness” before using them; summarize contributions with intuition and scope (not just math).

**Up-front definitions:** Define “reparameterization,” “short description length models,” and LVR/MLS/NMLS with a one-paragraph intuition each and where they are used.

**Section 3 reorganization:** Provide a roadmap; motivate each definition/lemma/proposition; introduce Eq. (7) with derivation/context, make its link to Def. 3.1 and Eq. (14) explicit; state assumptions near each result.

**Contributions list rewrite:** Replace math-only bullets with implications (“so what?”), scope, and limitations; make it accessible to a wider audience.
Figures/exposition: Add an explanatory figure contrasting sharp vs. flat minima and showing how LVR/MLS/NMLS behave on a toy landscape; captions must state takeaways.

---

> ### Author Response · Authors · 2025-11-26
> **Thank you**
>
> We thank Reviewer KMtE for the detailed feedback on presentation issues. We have undertaken a substantial rewrite of the paper to address these concerns.
>
> ### **Concern 1: Abstract Rewrite**
>
> > *"The abstract introduces 'compression' and 'reparameterization-invariant sharpness' with no setup or definitions, and the contribution bullets are stated as math without intuition..."*
>
> **Response:**
> We have completely rewritten the abstract with the following structure:
>
> 1. Clear problem statement (1-2 sentences opening)
> 2. Define "compression" in plain language before using the term
> 3. Define "reparameterization-invariant sharpness" before using it
> 4. Rewrite contribution bullets to include:
>    - What we did (method)
>    - Why it matters (implications)
>    - Scope and limitations
> 5. Remove math-only statements and add intuition throughout
>
> ### **Concern 2: Up-front Definitions**
>
> > *"Core terms ('reparameterization,' 'short description length models') aren't defined up front."*
>
> **Response:**
> We have added a prominent "Key terminology" section at the end of the introduction (after the roadmap) that provides:
>
> 1. **Reparameterization:** Intuitive explanation with example
> 2. **One-paragraph intuition for LVR, MLS, NMLS:** What each measures and why it matters
> 3. **OOD definition:** Added inline when first mentioned in Section 5
>
> Note: We removed "short description length models" as it was already explained inline where needed.
>
> The introduction now has two highlighted paragraphs for easy reference:
> - **"Paper organization"** (roadmap)
> - **"Key terminology"** (definitions)
>
> ### **Concern 3: Section 3 Reorganization**
>
> > *"Section 3... is poorly organized: definitions/lemmas/propositions arrive without motivation; Eq. (7) appears abruptly... and its link to Def. 3.1 is unclear."*
>
> **Response:**
> We have systematically reorganized Section 3 with the following improvements:
>
> 1. **Added "Key assumptions" paragraph** at start of Section 3 stating:
>    - Approximate interpolation solution
>    - Architecture (linear layers + nonlinear activations)
>    - Quadratic loss function
>    - Note about local regime and first-order Taylor approximations
>
> 2. **Added motivation before each definition/proposition:**
>    - Before Def 3.1 (LVR): Why volume compression measures representation compression
>    - Before Prop 3.4: States this is "first main result" linking sharpness to compression
>    - Before Def 3.2 (NVR): Why network-wide version yields tighter bounds
>    - Before Prop 3.6: Applying analysis to all layers
>    - Before Def 3.3 (MLS): Why maximum eigenvalue avoids product-of-eigenvalues issues
>    - Before Prop 3.8: Mirrors LVR bound but avoids product issue
>    - Before Def 3.5 (NMLS): Why network-wide version provides more consistent predictions
>    - Before Prop 3.10: Extending MLS analysis to all layers
>    - Before subsection 3.4: Why dimensionality measures uniformity of compression
>
> 3. **Clarified Equation 7 and its connections:**
>    - Added context: references change of variables formula from multivariate calculus
>    - Explained derivation: volume scaling given by Jacobian determinant
>    - Clarified formula for case where N < M
>    - Removed confusing forward reference to C_f^lim
>    - Added smooth transition before Def 3.1
>
> 4. **Highlighted introduction roadmap** with "Paper organization" heading to make it more prominent (decided against duplicating roadmap in Section 3 as it would be redundant)
>
> ### **Concern 4: Contributions List Rewrite**
>
> > *"The contribution list is math-only with no intuition, scope, or implications."*
>
> **Response:**
> We have completely rewritten the contributions list (Page 2) with each item now including:
>
> - **Bold headers** for clarity
> - **What was done** + **why it matters** + **scope of applicability**
> - Accessible language for wider audience
> - Dedicated "On bound tightness" paragraph with bold header after the list
>
> ### **Concern 5: Figures/Exposition**
>
> > *"Add an explanatory figure contrasting sharp vs. flat minima and showing how LVR/MLS/NMLS behave..."*
>
> **Response:**
> We acknowledge this would strengthen the paper. However, given the two-week rebuttal timeline and the substantial effort required to create new explanatory figures, we have not included it in this revision but consider it a valuable direction for future rounds of revision.

---

### Review · Reviewer_8k2n · 2025-11-11

**Summary Of Contributions:**

The current paper aims to build a connection between the sharpness quantity and quantities related to compression, both theoretically and
empirically.

It is a bit surprising that all the theoretical results are not that technical, and only some basic calculus and norm inequalities are used.

**Audience:**

Yes

**Audience Explanation:**

The author considers LVR and MLR, which quantify compression, and shows they are correlated with sharpness, both theoretically and empirically.

Though the bounds may not be that satisfactory, I would say the connection itself is interesting.

The author defers the discussion of the role of sharpness to generalization to the end; I do not think I really understand what the author is trying to say there. I will explain my confusion in more detail below.

In summary, I do not think the paper gives a theoretically founded and clear explanation of the relationship between sharpness and generalization. But raising the connection between sharpness and the two compression quantification measures is interesting.

**Broader Impact Concerns:**

I do not see issuses concerning ethical implications of the work.

**Claims And Evidence:**

Yes

**Claims Explanation:**

The connection between loss of flatness to compressed neural representations is explained using the bounds established in Section 3. I look over the experiments in Section 5, and I think the correlation is pretty clear for the specific model and data set.

However, I am not sure whether the connection is actually "SIMPLE." I will explain this in more detail later.

**Requested Changes:**

**"Simple" Connection:**

I am not sure whether the bounds given in Propositions 3.4, 3.6, 3.8, and 3.10 actually show a "simple " connection. At best, they are just simple bounds. For two networks with the same architecture and the same inputs, but with different weights, it is possible, based solely on the bounds, that the RHS is larger for one network, while the LHS is smaller for the same network.

Moreover, even ignoring the previous issue, those prefactors in those bounds are not controlled in any way, particularly for those inputs to the intermediate layers, for which the author needs some lower bounds. Isn't it very possible that those prefactors are very large, making the bound totally useless? For two networks with the same architecture and the same inputs, but with different weights, even if their sharpnesses are the same, the bounds can still be different because of the prefactors.

I think Reviewer sf7P has a similar concern.

So, I feel that the simple connection mentioned in the title is not well-grounded in terms of those theoretical bounds. At the very least, the author should provide more information about those bounds in their summary of contributions on page 2. And they shall explain a bit about the insufficiency of the bounds in a separate section, or a remark environment, or a paragraph starting with bolded font words indicating the insufficiency.

**Compression**:

A major contribution of the paper is the introduction of two compression measures: LVR and MLS. I am actually not entirely sure why these two quantities measure the compression. Can the author say a bit more? Also, should these quantities be less than 1 so that we know there is an actual "compression" going on? Do the bounds with sharpness (in Section 3) by any means show that those quantities can be less than 1, so some compression is going on?

**Claims and explanation I do not understand**

There are a few places where I do not understand what the author means. I hope the author can spend some time explaining them to me:

- On page 3, I do not understand the implication in the "therefore" in "We note that for the cross-entropy loss function, the Hessian vanishes as the cross-entropy (CE) loss approaches 0 (Granziol, 2020; Wu et al., 2018a). Therefore, the sharpness of CE loss cannot differentiate between local minima with different traces of the Hessian."

     To be more specific, I thought sharpness is the traces of the Hessian, so why can it not distinguish the local minima with different traces of the Hessian?

- Continuing the same paragraph, in the end, it says "1) MSE loss and CE loss share the same minimum, so Equation (3) will hold for networks trained with CE loss;"

    I honestly don't understand this. First, why can MSE and CE loss have the same minimum? Is there a mathematical proof of this? Second, why can Equation (3) still hold for the CE loss? I thought (3) is derived using the particular form of MSE, and such a form is not present for CE. In particular, I thought the loss cannot be 0 for the CE loss.

    At the very least, what are "our results" referred to here? The bounds in Section 3?

- In Section 5, the author says, "Sharpness alone is only directly responsible for the robustness of representations, and only together with other conditions or implicit biases does sharpness lead to superior generalization."

    I am not sure how compression relates to the robustness of representations. And I thought implicit biases seem to be too general. In the context of sharpness, I thought implicit biases meant biasing towards flat solutions.

- In Section 5, the second paragraph, the author says, "..., but Wortsman et al. (2022) shows that a large learning rate can severely hurt OOD generalization performance. More intuitively, consider a scenario where one provides a large language model (LLM) with a long text sequence and instructs it to find a specific piece of information, often called the "needle in a haystack" test. Even a slight alteration in the instructions (a tiny portion of the input) given to the model should lead to a notable difference in its output, depending on the desired information. Therefore, compression of network output is not a desirable
property in this case."

    Two things:
    - First, what is OOD?

    - Second, I do not understand the example here. I thought the network was compressed, but the output is not compressed, at least not directly. What do you mean by the compression of network output? It sounds like the example is basically saying the input-output relationship is very delicate, and we cannot use a compressed or simple network to model such a delicate relationship.

- At the beginning of Section 3.5, the author said, "(though extensions to network versions, that treat robustness to features that develop earlier in the network may be possible and are a focus of ongoing study)." This sentence does not look correct. Please consider rewriting it.

- I do not understand why the author says, "with high probability" before (13). I thought (13) is a deterministic result. Since you only need (14), can you give a more rigorous derivation of the last equality of (14)?

**Other comments**

- It is probably important to note that everything here has no RELU.

- Please mention where the proof of Lemma 3.2 is.

- In end of Page 6, the $C_{B ( \bar x)}$ seems to be weird

---

> ### Author Response · Authors · 2025-11-26
> **Thank you**
>
> We thank Reviewer 8k2n for the positive assessment and feedback.
>
> ### **Concern 1: "Simple" Connection and Bound Tightness**
>
> > *"I am not sure whether the bounds... actually show a 'simple' connection... those prefactors... are not controlled in any way... provide more information about those bounds in [the] summary of contributions."*
>
> **Response:**
>
> 1. **"On bound tightness" paragraph (Page 2):** Acknowledges bounds may not be tight, depend on weight norms/input magnitudes, and require empirical validation.
>
> 2. **Enhanced contribution bullets (Page 2):** Each now includes what was done, implications, scope, and limitations.
>
> 3. **"On weight norms in bounds" paragraph (Section 3):** Explains lower sharpness doesn't always guarantee more compression due to weight norm terms; distinguishes single architecture (stable prefactors) vs. across architectures (adaptive normalization needed); references Appendix H.
>
> ### **Concern 2: Why LVR/MLS Measure Compression**
>
> > *"I am... not entirely sure why these two quantities measure the compression. Can the author say a bit more? Also, should these quantities be less than 1 [for] actual 'compression'?"*
>
> **Response:**
> Added paragraph "Why LVR and MLS quantify compression" (Section 3):
>
> - **Compression:** Concentration/contraction under local perturbations
> - **LVR:** Volume contraction via Jacobian; < 1 = compression, = 1 = preservation, > 1 = expansion
> - **MLS:** Reduced sensitivity; lower = nearby inputs → nearby outputs
> - **Note:** Bounds don't guarantee LVR < 1, but empirically compression occurs (Section 5)
>
> ### **Concern 3: Cross-Entropy Loss Discussion (Page 3)**
>
> > *"I do not understand the implication in the 'therefore'... why can [sharpness] not distinguish... local minima with different traces...? Why can MSE and CE loss have the same minimum? ... why can Equation (3) still hold for the CE loss?"*
>
> **Response:**
> Revised paragraph (Section 2, "Scope and loss function considerations"):
>
> 1. **States:** Theoretical analysis uses MSE loss
> 2. **Corrects CE discussion:** Per Granziol et al., large weights + low CE loss → artificially small Hessian (deceptively flat)
> 3. **Explains issue:** Hessian rank bounded by (neurons × classes) causes inherent zero eigenvalues
> 4. **Removes claim:** Incorrect statement about MSE/CE sharing minimum removed
> 5. **Two pathways for CE-trained networks:** (a) Extension via logistic loss with label smoothing; (b) Computing metrics post-hoc using MSE sharpness (ViT experiments)
>
> ### **Concern 4: Section 5 Unclear Statements**
>
> > *"I am not sure how compression relates to the robustness of representations... implicit biases seem... too general... I do not understand the example... What do you mean by the compression of network output?"*
>
> **Response:**
>
> 1. **Defined "robustness of representations":** Small input changes → small feature changes (quantified by low LVR/MLS)
>
> 2. **Replaced "implicit biases"** with "other latent factors"
>
> 3. **Fixed needle-in-haystack example:** Now distinguishes robustness to random perturbations (what metrics measure) vs. sensitivity to meaningful changes (what tasks require); clarified when compression helps/hurts generalization
>
> 4. **Added "OOD" definition** inline
>
> ### **Concern 5: Section 3.5 Unclear Sentence**
>
> > *"...(though extensions to network versions, that treat robustness to features that develop earlier...)." This sentence does not look correct.*
>
> **Response:**
> Removed confusing sentence and rewrote neural collapse discussion:
> - **Focuses on math:** bound effective when K ≥ d
> - **Application domains:** language modeling, retrieval, face recognition
> - **Within-class samples connection:** Viewed as perturbations around class mean. When within-class variance vanishes, perturbations produce minimal feature variation—maximal compression captured by LVR/MLS.
> - **Caveat:** "the precise relationship between our local robustness bounds and... global... structure of neural collapse... remains an open question"
>
> ### **Concern 6: "With high probability" before Eq. 13**
>
> > *"I do not understand why the author says 'with high probability' before (13). I thought (13) is... deterministic."*
>
> **Response:**
> Correct. Removed "with high probability" from line 309 and changed to "in the limit α → 0" since the Taylor expansion is deterministic.
>
> ### **Concern 7: Technical Issues**
>
> > *"It is probably important to note that everything here has no RELU."*
>
> **Response:**
> Stated in Section 3 "Key assumptions" paragraph: "we do not assume specific activation functions for the main bounds." Theory applies to general smooth nonlinearities and extends to residual connections (Appendix B).
>
> > *"Please mention where the proof of Lemma 3.2 is."*
>
> **Response:**
> Clarified that the proof follows directly from AM-GM inequality and is thereby omitted.
>
> > *"In end of Page 6, the [symbol] seems to be weird"*
>
> **Response:**
> Fixed the symbol error.

---

### Decision · Action_Editor_Ae9y · 2025-12-28

**Recommendation:** Accept as is

**Audience:**

Yes

**Audience Explanation:**

Most reviewers find the central connection (loss landscape geometry vs. representation geometry) as timely, insightful, and of broad interest.

**Claims And Evidence:**

Yes

**Claims Explanation:**

Based on that assessment, the submission’s claims appear to be supported by evidence that is accurate, convincing, and reasonably clear.

After the rebuttal, most reviewers explicitly state the theoretical results ``appear technically sound and correct,'' and the arguments are checkable with few obscure steps.